# OPERATOR FLOW MATCHING FOR TIMESERIES FORECASTING

## ABSTRACT

Forecasting high-dimensional, PDE-governed dynamics remains a core challenge for generative modeling. Existing autoregressive and diffusion-based approaches often suffer cumulative errors and discretisation artifacts that limit long, physically consistent forecasts. Flow matching offers a natural alternative, enabling efficient, deterministic sampling. We propose TempO, a time-conditioned latent flow matching method mimicking classical PDE evolution operators. We introduce an attention-based multiscale autoencoder, a latent Fourier vector field regressor, and decoupled spatial and temporal processing for temporally coherent and accurate rollouts. We prove an upper bound on FNO approximation error and empirically show that TempO outperforms state-of-the-art baselines across three benchmark PDE datasets, and spectral analysis further demonstrates superior recovery of multi-scale dynamics, while efficiency studies highlight its parameter- and memory-light design compared to attention-based or convolutional regressors.

## 1 INTRODUCTION

Generative artificial intelligence has brought unparalleled creative and scientific potential, with models capable of producing images (Hatamizadeh et al., 2025), video (Bar-Tal et al., 2024), audio (Ju et al., 2024), and text (Grattafiori et al., 2024) that rival human quality. From autoregressive transformers to diffusion models and energy-based approaches, the landscape of generative AI is rich and diverse, offering multiple pathways to model complex data distributions. At the core of this revolution are probabilistic generative models, which learn to sample from complex, high-dimensional distributions. Among these, flow matching models have emerged as a class of generative models which learn to transform a simple prior distribution to a more complex data distribution as a continuous transformation. This direct, simulation-free approach enables both efficiency and precision, offering a new lens on modeling complex systems (Lipman et al., 2023).

Despite recent advances, forecasting high-dimensional temporal dynamics remains challenging. Deep learning models are computationally expensive and often fail catastrophically after a few dozen timesteps due to compounding errors in autoregressive predictions, particularly in stiff or chaotic systems (Ansari et al., 2024; Raissi et al., 2019). Even with the advent of large language models and their remarkable ability to generate, models that attempt to leverage them for forecasting introduce truncation and quantisation errors due to tokenised representations which further exacerbate cumulative errors (Ansari et al., 2024), offering little practical benefit relative to their computational cost (Tan et al., 2024). Modern generative models have been proven capable of generating visually compelling and coherent videos (John et al., 2024), but critically lack the fine-grained control required to be used in scientific and engineering contexts.

Recent foundation models for forecasting include GenCast for weather (Price et al., 2025) and Chronos for general time series (Ansari et al., 2024), demonstrate the promise of large-scale pretraining. These models leverage massive datasets across multiple domains resulting in strong zero- and few-shot transferability. Chronos captures coarse, long-range correlations remarkably long timespans; however, the granularity, i.e. prediction length still falls at an average of 22 across 55 datasets, with only 7 tasks exceeding 30 steps (Ansari et al., 2024). GenCast, likewise, can generate 15-day global weather forecasts, but at a granularity of 12 hours, around 30 steps. True progress requires models capable of deterministic yet flexible generation, able to explore plausible trajectories while respecting physical constraints to then select precise forecasts out of the space of plausible predic-

tions (Guo et al., 2025). Although the short to mid term range is a popular horizon to explore (Lim et al., 2025b), the goal is to generate long-horizon predictions on the order of 30 timesteps or more, generating trajectories that are not just plausible, but physically consistent.

Fundamentally, models relying on discretisation or tokenisation are not ideal for continuous, Partial Differential Equation (PDE)-governed dynamics. Demonstrating smooth trajectories in state space which generalise to long forecasting horizons would show greater fidelity to the underlying physics. Other existing efforts which leverage diffusion (Molinaro et al., 2025; Yao et al., 2025; Huang et al., 2024) move toward more natural representations and, in some settings, benefit from the stochasticity inherent to diffusion, e.g. for uncertainty quantification or modeling ensembles of plausible trajectories in chaotic regimes (Lippe et al., 2023). However, their stochastic sampling procedures can introduce computational overhead and variance when the goal is to learn a sharp, deterministic operator. Flow matching provides a complementary alternative: vector field regression aligns naturally with learning PDE operators, which themselves describe deterministic time derivatives, and yields efficient Ordinary Differential Equation (ODE)-based sampling without iterative denoising. Existing flow matching methods have individually worked toward video generation (Davtyan et al., 2023; Jin et al., 2025; Holzschuh et al., 2025; Serrano et al., 2024) and PDE single-step prediction (Kerrigan et al., 2023), but thus far have not been thoroughly tested for long-horizon temporal forecasting and do not design for the deterministic and stable rollouts required for such tasks.

In this work, we propose Temporal Operator flow matching (TempO), the first principled integration of time-conditioned latent flow matching with neural operators mimicking classical PDE evolution operators, offering an efficient, stable and spectrally accurate alternative to autoregressive and diffusion-based PDE forecasting. TempO is built around four key innovations:

1. We design a multi-headed attention–enhanced autoencoder that learns multiscale latent representations, allowing the flow field to jointly capture global structures and fine local features.

2. We construct time-conditioned latent spectral embeddings that encode PDE states in a frequency-aware latent space, enabling smooth flow interpolation and preserving high-frequency dynamics.

3. Motivated by the structure of PDE evolution operators, TempO explicitly decouples spatial and temporal processing via channel folding, using operator layers for spatial modes and latent flow dynamics for temporal evolution.

4. We incorporate sparse conditioning, providing computational and data efficiency by conditioning the flow only on two prior timesteps without degrading accuracy.

We derive theoretical error bounds that characterize the efficiency and expressivity of TempO, and showcase its performance on PDE benchmarking datasets accompanied with a spectral analysis showing a distinct advantage in capturing the essential dynamics required for forecasting. Empirically, we see a 16% lower error when predicting vorticity of 2D incompressible Navier Stokes, with Pearson correlations remaining above 0.95 for a 40 step forecasting horizon, demonstrating its stable temporal forecasting and high quality generation capability.

## 2 RELEVANT WORKS

Application-specific models for scientific data have also seen development: GenCFD (Molinaro et al., 2025) proposes a conditional diffusion model to generate the underlying distributions of high fidelity flow fields. Kerrigan et al. (2023) propose the first extension of Fourier Neural Operators (FNOs) to flow matching tasks and predicts plausible fluid dynamic fields, Functional Flow Matching. Shi et al. (2025) builds on this concept and extends it to learning stochastic process priors on function spaces. Similarly, Lim et al. (2025a) extend denoising diffusion models to function space, introducing Denoising Diffusion Operators for unconditional prior generation; Yao et al. (2025) extends this to conditional posterior sampling under observation constraints and sees state-of-the-art performance for multi-resolution PDE tasks, as compared to its competitor DiffusionPDE Huang et al. (2024) which originally demonstrated strong performance in solving PDEs with partial observations. Such methods have thus far focused on static prediction, i.e., generating diverse samples of plausible PDE solutions, rather than deterministic temporal rollouts.

Models designed to predict sequences of future states include the aforementioned large-scale Chronos and GenCast (Ansari et al., 2024; Price et al., 2025). In addition, pyramidal flow matching (Jin et al., 2025) produces state-of-the-art video generation compared to leading models (Zheng et al., 2024), representing a successful flow matching foundation model. (Tamir et al., 2024) present conditional flow matching for time series, succeeding in long 1D trajectories where neural ODEs fail, but has not scaled to 2D spatiotemporal data. Physics-Based Flow Matching (Baldan et al., 2025) adds a PINN-style loss (Raissi et al., 2019) to flow matching for surrogate model. Kollovieh et al. (2024) extends this with Gaussian processes for forecasting tasks outside of scientific machine learning. We focus instead on models that fall between these two categories, scaling reasonably to 2D data to match common PDE settings. Li et al. (2025) further leverage latent space modeling and coarsely sampled diffusion for PDE generation on irregular grids.

TempO occupies a distinct methodological position in this landscape: it is conditional and deterministic, built to learn an operator-valued transport that maps an initial PDE state to its evolved state, rather than to sample diverse static solutions. TempO also shares the same spectral motivation as prior works, e.g. Functional Flow Matching, but differentiates itself in a crucial way: instead of using an FNO as a denoiser inside a sampler, TempO uses an FNO to parameterize a latent-space velocity field that is time-conditioned and integrated as an ODE. This design enables deterministic, operator-valued rollouts rather than stochastic sample generation, and directly targets long-horizon forecasting stability.

## 3 METHOD

We begin by developing the background which is then used to construct our method. Flow matching learns a time-dependent velocity field $v_\theta(z, t)$ defining an ODE in the latent space:

$$\frac{dz(t)}{dt} = v_\theta(z(t), t), \quad z(0) \sim \pi_0, \tag{1}$$

where $\pi_0$ is a simple prior (e.g., Gaussian). Integrating this ODE transports samples to the latent data distribution $\pi_1$, see Appendix B. Training reduces to a regression objective that matches the model velocity field to a target velocity along interpolating probability paths (Lipman et al., 2023). This enables simulation-free sampling from complex distributions.

Table 1: Representative Path Choices in Flow Matching Models.

| Path | $a_t$ | $b_t$ | $c_t$ | Parameter definitions |
|---|---|---|---|---|
| Affine-OT[1] | $t$ | $0$ | $(1 - (1 - \epsilon_{\min})t)^2$ | $\epsilon_{\min} \geq 0$: min. noise level |
| RIVER[2] | $(1 - (1 - \sigma_{\min})t)$ | $t$ | $\sigma^2$ | $\sigma \geq 0$: noise scale, $\sigma_{\min} \geq 0$: min. noise |
| SLP[3] | $(1 - t)$ | $t$ | $\sigma_{\min}^2 + \sigma^2 t(1 - t)$ | $\sigma, \sigma_{\min} \geq 0$: variance parameters |
| VE-diff[4] | $1$ | $0$ | $\sigma_t^2$ | $\sigma_t$: geometric schedule, $\sigma_{\min}, \sigma_{\max} > 0$ |
| VP-diff[4] | $\exp(-\frac{1}{2}T(1 - t))$ | $0$ | $1 - \exp(-T(1 - t))$ | $\beta_{\min}, \beta_{\max} > 0, T(t) = \int_0^t \beta(s)\, ds$ |

[1] (Lipman et al., 2023), [2] (Davtyan et al., 2023), [3](Lim et al., 2025b),[4] (Ryzhakov et al., 2024)

A key component of flow matching is the choice of the probability density path $p_t$ interpolating between the reference distribution $\pi_0$ and the target $\pi_1$. We focus on Gaussian conditional paths with closed-form velocity fields:

$$p_t\Big(Z \mid \tilde{Z} := (Z_0, Z_1)\Big) = \mathcal{N}\Big(Z \mid a_t Z_0 + b_t Z_1,\ c_t^2 I\Big),$$

where $a_t, b_t, c_t$ define the path (Table 1). This pair-conditional path is defined for a specific transition $(Z_0, Z_1)$, and the marginal interpolant is obtained by averaging over all pairs: $p_t(Z) = \mathbb{E}_{(Z_0, Z_1)}[p_t(Z \mid Z_0, Z_1)]$. While $\pi_0$ is typically a standard Gaussian, intermediate densities $p_t$ can follow diffusion-inspired, optimal transport, or other custom schedules.

To parameterize $v_\theta$, we modify FNOs, which approximate mappings between functions via spectral convolution layers. Given input $u$, the FNO parameterizes an operator as $\mathcal{G}_\theta : u \mapsto \tilde{u}, \quad \tilde{u} : \mathcal{D} \to \mathbb{R}^{c_{\text{out}}}$, that maps $u$ to an output function $\tilde{u}$. Iterative Fourier layers perform spectral transformations of the input $\hat{u}(k) = \mathcal{F}[u](k), \quad \hat{\tilde{u}}(k) = R_\theta(k) \cdot \hat{u}(k)$, followed by an inverse Fourier transform back to the spatial domain; $\tilde{u}(x) = \mathcal{F}^{-1}[\hat{\tilde{u}}](x)$, with $R_\theta(k)$ being learnable Fourier-mode weights and $\mathcal{F}$ denoting the Fourier transform. This spectral representation allows the FNO to efficiently capture long-range dependencies and global correlations in the data.

## 3.1 Temporal Operator flow matching (TempO)

Using an FNO-inspired regressor to learn the vector field of a flow matching model has a number of benefits, namely, the added expressivity that the Fourier representation provides at a low computational cost thanks to highly optimised Fast Fourier Transform (FFT) operations. Building on prior analysis of FNOs for solving PDEs (Kovachki et al., 2021), we show that an FNO-inspired regressor can achieve an upper bound on approximation error for flow matching models and we provide a lower bound on the accuracy achievable by sampler-based methods (e.g., Transformer or U-Net) in relation to their number of parameters.

**Theorem 3.1** (FNO regressor constructive upper bound). *Let $\mathbb{T}^d$ be the $d$–torus. Fix $s, s' \geq 0$ and let $\mathcal{U} \subset H^s(\mathbb{T}^d)$ be compact. Suppose $\mathcal{G} : \mathcal{U} \to H^{s'}(\mathbb{T}^d)$ is continuous and satisfies $|\widehat{\mathcal{G}(u)}(k)| \leq C_\lambda(1 + |k|)^{-p}$ for all $u \in \mathcal{U}$, $k \in \mathbb{Z}^d$, with constants $C_\lambda > 0$, $p > 0$. If $p > s' + \frac{d}{2}$ and we define $\alpha := p - s' - \frac{d}{2} > 0$, then for every $\varepsilon > 0$ there exists a Fourier Neural Operator $\mathcal{G}_\theta$ with*

$$P_{FNO}(\varepsilon) \lesssim \varepsilon^{-d/\alpha},$$

*such that $\sup_{u \in \mathcal{U}} \|\mathcal{G}(u) - \mathcal{G}_\theta(u)\|_{H^{s'}} \leq \varepsilon$. The hidden constants depend only on $d, s, s', \mathcal{U}, C_\lambda$ and mild/logarithmic factors.*

This result is in line with the estimates and arguments made in (Kovachki et al., 2021).

*Sketch of proof of Theorem 3.1.* (Spectral truncation.) The Fourier decay assumption implies that high-frequency modes of $\mathcal{G}(u)$ contribute at most $O(K^{-2\alpha})$ to the $H^{s'}$-error. Choosing $K \asymp \varepsilon^{-1/\alpha}$ makes this truncation error $\leq \varepsilon/2$.

(Finite-dimensional reduction.) For this cutoff $K$, the operator $\mathcal{G}_K$ is determined by $O(K^d)$ Fourier coefficients, and inputs can likewise be restricted to finitely many low modes without significant loss of accuracy. Thus the problem reduces to approximating a continuous map between compact subsets of $\mathbb{R}^{m_{in}}$ and $\mathbb{R}^{m_{out}}$, with $m_{out} \asymp K^d$.

(Approximation by networks.) Standard universal approximation results (or the constructive FNO design in (Kovachki et al., 2021)) ensure that such a finite map can be uniformly approximated by a network with $O(K^d)$ parameters, up to mild logarithmic factors.

(Conclusion.) Combining these errors yields an overall accuracy $\varepsilon$ with parameter count $P \lesssim K^d \asymp \varepsilon^{-d/\alpha}$, proving the claim. $\qquad\square$

**Proposition 3.2** (Transformer/UNet Sampler-based lower bound). *Under the assumptions of Theorem 3.1, consider any learner that observes each $u \in \mathcal{U}$ only through $n$ fixed point evaluations and applies a parametric map with $P$ real parameters, required in the worst case to reconstruct all Fourier modes up to radius $K \asymp \varepsilon^{-1/\alpha}$. Then necessarily*

$$n \gtrsim \varepsilon^{-d/\alpha}, \qquad P_{sampler}(\varepsilon) \gtrsim \varepsilon^{-\beta d/\alpha},$$

*for some architecture–dependent $\beta \geq 1$ (optimistically $\beta = 1$ when only diagonal mode-wise maps are needed, generically $\beta = 2$ for arbitrary dense linear maps). These bounds are information-theoretic and asymptotic, up to constants and mild/logarithmic factors.*

*Sketch of proof of Proposition 3.2.* (Sampling necessity.) The $K$–mode subspace $V_K$ has dimension $D_K \asymp K^d$. Sampling at $n$ points defines a linear map $S : V_K \to \mathbb{C}^n$. For $S$ to be injective on $V_K$, its matrix must have rank $D_K$, hence $n \geq D_K \asymp K^d$.

(Parameter complexity.) After sampling, the learner implements a parametric map $M : \mathbb{C}^n \to \mathbb{C}^m$. To represent arbitrary linear maps on the $D_K$-dimensional coefficient space (e.g. arbitrary diagonal multipliers), the parameter family must have at least $P \gtrsim D_K$ degrees of freedom. For fully general dense linear maps one needs $P \gtrsim D_K^2$.

(Conversion.) Substituting $K \asymp \varepsilon^{-1/\alpha}$ (from the theorem) gives $n \gtrsim \varepsilon^{-d/\alpha}$ and $P \gtrsim \varepsilon^{-\beta d/\alpha}$ with $\beta = 1$ (optimistic) or $\beta = 2$ (dense case), establishing the lower bound, see Appendix A for the extended proof. $\qquad\square$

**Corollary 3.3** (FNO vs sampler scaling). *From Theorem 3.1 and Proposition 3.2 one has*

$$P_{\mathrm{FNO}}(\varepsilon) \lesssim \varepsilon^{-d/\alpha}, \qquad P_{\mathrm{sampler}}(\varepsilon) \gtrsim \varepsilon^{-\beta d/\alpha}.$$

*Hence, whenever $\beta > 1$, FNOs achieve the same accuracy $\varepsilon$ with asymptotically fewer parameters than sampler–based learners.*

**TempO**   Consequently, we propose TempO, a framework capable of long rollout PDE forecasting via a multiscale attention-based autoencoder, time-conditioned FNO vector field regressor, and channel folding for both efficiency and enhanced temporal coherency, reflecting the PDE evolution-operator perspective that motivates our decoupling of spatial and temporal processing. Together with temporal conditioning, these define a novel, end-to-end trainable model for predicting latent dynamics.

Let $f_\phi : \mathbb{R}^X \to \mathbb{R}^Z$ denote an encoder mapping data points $x$ to latent embeddings $z = f_\phi(x)$. We use a *multi-headed attention-enhanced autoencoder* for multiscale embeddings that are well-suited for subsequent temporal conditioning and flow-based evolution. The use of attention layers and residual blocks preserves multi-scale spatial correlations while compressing the data for more efficient processing. We can then define a latent-space velocity field described by 1 where $v_\theta$ is parameterized by an FNO, which provides a spectral inductive bias for learning PDE-consistent dynamics.

Under the standard flow-matching assumption that the learned drift has bounded Lipschitz constant, the global forecasting error grows no faster than $\exp\left(\int_0^1 L_t \, dt\right) \varepsilon$ yielding polynomial accumulation, consistent with classical ODE stability. In contrast, autoregressive models accumulate errors through the product of stepwise Jacobians, $\prod_t \|J_t\|$, which can increase much more rapidly. We further mitigate the expected error accumulation via sparse conditioning.

*Sparse conditioning* provides computational and data efficiency by leveraging the locality and temporal coherence inherent to PDEs, and additionally allows the model to condition on fixed points during rollout rather than recursively using its own predictions as inputs (Davtyan et al., 2023; Lim et al., 2025b). For some discrete-time sequence $\{x_t\}_{t=1}^N$ with $x_t \in \mathcal{X}$, its latent representation is given by $\{z_t\}_{t=1}^N$, where $z_t = f_\phi(x_t)$. For a prediction horizon $T \in \{L, \ldots, N-1\}$ with sequence length $L$, the objective is to predict the next latent embedding $z_{T+1}$. We define a reference embedding to be $z_T$, corresponding to the most recent observation prior to the prediction target, and a conditioning embedding as some observation selected at a timestep $\tau \in \{T-L, \ldots, T-1\}$. These two embeddings are concatenated with the temporal offset, defined as $\Delta = T - \tau$, which is the extent of temporal data the model is provided to predict the next-step embedding, $\hat{z}_{T+1} = f_\theta(z_T, z_\tau, \Delta)$. In practice, we fix the conditioning to one of the provided initial steps, effectively pinning the generation against a known correct solution and incrementing the temporal offset and updating the reference embedding: this results in significantly more stable rollouts.

We then propose *channel folding*, a key architectural contribution that allows 2D FNO layers to process time-varying latent fields without temporal blurring. This folding preserves a clean separation between spatial operator structure and temporal latent dynamics, reflecting the semigroup property of PDE evolution operators. We collapse the batch and channel axes into a single "effective batch" dimension $u' \in \mathbb{R}^{(B \cdot C) \times T \times H \times W}$ as input to the FNO. This folding operation effectively treats each channel of each sample as an independent element within the extended batch. As a consequence, the FNO is applied identically across all channels but without cross-channel mixing at this stage. This disentangles $t$ (within the flow matching integration step) from $\tau$ (the PDE timestep), which is explicitly provided as part of the conditioning tuple $\mathcal{C} = \{z_\tau, \Delta\}$.

This *time-conditioned FNO* then operates over latent temporal embeddings as functions on their spatial domain $v_\theta(z, t) = \mathcal{G}_\theta(z)$ to learn the time-dependent vector field that transports a reference latent distribution $\pi_0$ to the latent data distribution $\pi_1$. This differs from prior FNO-based generative models, which operate either unconditionally or autoregressively in input space rather than in a time-conditioned latent ODE. By leveraging the spectral inductive bias of FNOs, the learned velocity field models cross-scale correlations, thereby stabilizing flow matching across long horizons. Combined with spectral latent embeddings, this produces temporally coherent latent trajectories that maintain high-frequency fidelity during transport.

## 4 EXPERIMENTS

The TempO is evaluated with the goal of assessing its ability to learn accurate stochastic latent-space dynamics and forecast high-dimensional solution fields over medium to long time horizons. We test our method over PDE datasets which pose challenging spatio-temporal correlations and multiscale features, making them a natural testbed for latent flow-based modeling.

Our proposed TempO was set against five key methods. The state-of-the-art video generation method based on a U-Net shaped Vision Transformer (ViT) and modified optimal transport path Random frame conditioned flow Integration for VidEo pRediction (RIVER) proposed by Davtyan et al. (2023) matches or surpasses common video prediction benchmarks using 10x fewer computational resources (Davtyan et al., 2023). We also include the baseline conditional flow matching Lipman et al. (2023) which implements a U-Net trained using a theoretically optimal affine optimal transport (Affine-OT) path. The stochastic linear path (SLP) was proposed by Lim et al. (2025b), tested with a ViT to directly address the challenges of spatiotemporal forecasting for PDE datasets. The Transformer-based latent space flow matching method with Affine-OT proposed by Dao et al. (2023) further demonstrates competitive performance in image generation using latent flow matching compared against both flow matching models and diffusion models (Phung et al., 2023; Ho et al., 2020) among others. We also evaluate both variance preserving diffusion (VP-diff) and variance exploding diffusion (VE-diff) paths which generalise the Denoising Diffusion Probabilistic noise perturbation model and a score-based model to flow matching paths, respectively (Ho et al., 2020; Song et al., 2021). Ryzhakov et al. (2024) establishes strong theoretical backing for both paths.

We then ablate the specific implementation of the methods (consisting of a specific architecture and a specific probability path). In summary, the choice of regressor includes our proposed TempO regressor, and additionally implement a ViT regressor (Davtyan et al., 2023; Lim et al., 2025b) and a classic U-Net regressor (Lipman et al., 2023). We pretrain a convolutional autoencoder with residual and attention blocks to obtain a compressed latent representation of the dynamics, see Appendix D. We additionally compare against baseline, non-flow matching models FNO2D, FNO3D, Wavelet Neural Operator (WNO)2D, WNO3D, and a U-Net to contextualize performance, with further details in Appendix F (Li et al., 2021; Tripura & Chakraborty, 2022).

All flow matching methods were conditioned using sparse conditioning, and baseline 2D methods mapping 10 timesteps to predict the following, and 3D methods directly mapping the block of 10 timesteps to directly predict the remaining 40. Flow matching models are then supervised by each probability density paths described in Table 1. Further details in Appendix E. The Adam optimiser with a learning rate of 1e-4 was used for the FNO, and 5e-5 for the ViT and U-Net regressors. Models are trained on an 80/10/10 training to validation to test data splits.

We evaluate our models on three spatiotemporal PDE datasets: the shallow water equation (SWE), which simulate 2D free-surface flows; 2D reaction diffusion (RD-2D) exhibiting multiscale non-linear patterns; and 2D incompressible Navier-Stokes vorticity (NS-$\omega$) dataset capturing chaotic turbulent dynamics. During training, models are sparsely conditioned on the first 15 frames and tasked with predicting the subsequent frame at resolutions of $1 \times 128 \times 128$ (shallow water equation (SWE)), $2 \times 128 \times 128$ (2D reaction diffusion (RD-2D)), and $1 \times 64 \times 64$ (2D incompressible Navier-Stokes vorticity (NS-$\omega$)), see Appendix G.

## 5 RESULTS

Overall, TempO outperforms the methods proposed by Lim et al. (2025b); Song et al. (2022); Lipman et al. (2023) and Davtyan et al. (2023) as well as the ablated flow-matching methods. For results predicting NS-$\omega$ in Table 2, we observe a 16% improvement in MSE and an 11.4% lower spectral MSE, producing spatially and spectrally accurate next steps. Its lower RFNE indicates reduced scale-independent error, while SSIM shows improved fidelity in local features, critical for the localized vorticity patterns where small spatial distortions significantly affect downstream evolution (Majda & Bertozzi, 2001). PSNR and Pearson see lower normalised ranges in values, indicating that large scale features like the vorticity intensity and global structure agreement, respectively, are more easily captured across all models, with a clear advantage by TempO; additional visualisations in Appendix I. We observe that the baseline FNO models outperform across the board for next step, but critically fail at longer rollouts evidenced by the 58.1% worse MSE/time of FNO-3D versus

Table 2: NS-$\omega$ Results: Comparison of TempO, U-Net, and ViT models.

| Regressor | Path | MSE ↓ | SpectralMSE ↓ | RFNE ↓ | PSNR ↑ | Pearson ↑ | SSIM ↑ | MSE/time ↓ |
|---|---|---|---|---|---|---|---|---|
| TempO | RIVER | **5.63e-02** | **3.84e-02** | **2.50e-01** | **25.19** | **0.969** | 0.786 | **2.67e-02** |
| | Affine-OT | 5.77e-02 | 3.98e-02 | 2.54e-01 | 25.08 | 0.968 | **0.789** | 2.91e-02 |
| | VP-diff | 8.10e-02 | 5.34e-02 | 2.85e-01 | 23.61 | 0.955 | 0.731 | 2.29e-01 |
| | VE-diff | 2.96e-01 | 1.73e-01 | 5.60e-01 | 17.98 | 0.821 | 0.373 | 5.02e-01 |
| ViT | Affine-OT[1] | 6.75e-02 | 4.38e-02 | 2.72e-01 | 24.40 | 0.962 | 0.758 | 8.71e-02 |
| | RIVER[2] | 6.88e-02 | 4.33e-02 | 2.73e-01 | 24.32 | 0.962 | 0.750 | 3.85e-02 |
| | VP-diff[3] | 7.77e-02 | 4.95e-02 | 2.89e-01 | 23.79 | 0.956 | 0.729 | 6.65e-02 |
| | VE-diff[3] | 1.63e+00 | 9.27e-01 | 1.35e+00 | 10.67 | 0.118 | 0.024 | 1.67e+00 |
| U-Net | VP-diff[4] | 4.05e-01 | 3.26e-01 | 6.71e-01 | 16.62 | 0.756 | 0.323 | 2.66e-01 |
| | RIVER | 4.08e-01 | 3.28e-01 | 6.74e-01 | 16.59 | 0.752 | 0.321 | 2.79e-01 |
| | Affine-OT[5] | 4.10e-01 | 3.42e-01 | 6.76e-01 | 16.57 | 0.751 | 0.324 | 2.82e-01 |
| | VE-diff[4] | 5.02e-01 | 3.70e-01 | 7.48e-01 | 15.68 | 0.694 | 0.263 | 2.92e-01 |
| Baselines | FNO-2D | *6.09e-04* | *4.27e-04* | *2.54e-02* | *44.85* | *1.000* | *0.992* | 1.92e-01 |
| | FNO-3D | 1.06e-01 | 7.34e-02 | 3.37e-01 | 22.46 | 0.945 | 0.645 | 6.37e-02 |
| | WNO-2D | 3.72e-03 | 2.83e-03 | 6.06e-02 | 36.99 | 0.998 | 0.966 | 5.19e-01 |
| | WNO-3D | 2.23e-01 | 1.19e-01 | 4.97e-01 | 19.209 | 0.868 | 0.452 | 2.05e-01 |
| | U-Net | 2.47e-03 | 1.92e-03 | 4.83e-02 | 38.772 | 0.999 | 0.976 | 1.66e+00 |

[1] (Dao et al., 2023), [2](Davtyan et al., 2023), [3] (Lim et al., 2025b; Song & Ermon, 2020), [4](Ryzhakov et al., 2024), [5](Lipman et al., 2023)

Table 3: SWE and RD-2D Results: Comparison of TempO, U-Net, and ViT models.

| Dataset | Method | MSE ↓ | SpectralMSE ↓ | RFNE ↓ | PSNR ↑ | Pearson ↑ | SSIM ↑ | MSE/time ↓ |
|---|---|---|---|---|---|---|---|---|
| SWE | TempO$_{\text{Affine-OT}}$ | **6.64e-05** | **5.65e-05** | **7.64e-03** | **46.5** | **0.998** | **0.997** | **1.60e-03** |
| | ViT$_{\text{Affine-OT}}$[1] | 9.59e-05 | 7.93e-05 | 9.06e-03 | 44.9 | 0.997 | 0.995 | 7.02e-03 |
| | ViT$_{\text{VP-diff}}$[2] | 1.30e-04 | 8.81e-05 | 1.05e-02 | 43.6 | 0.996 | 0.993 | 1.61e-03 |
| | ViT$_{\text{RIVER}}$[3] | 2.99e-04 | 1.67e-04 | 1.63e-02 | 40.0 | 0.992 | 0.981 | 6.96e-03 |
| | ViT$_{\text{SLP}}$[4] | 6.60e-04 | - | 1.28e-01 | 36.1 | - | 0.93 | - |
| RD-2D | TempO$_{\text{Affine-OT}}$ | **2.76e-05** | **2.18e-05** | **3.29e-02** | **65.7** | **1.000** | 0.999 | **1.89e-02** |
| | U-Net$_{\text{Affine-OT}}$[5] | 3.09e-05 | 2.45e-05 | 3.57e-02 | 65.2 | 0.999 | **0.999** | 1.95e-02 |
| | ViT$_{\text{Affine-OT}}$ | 6.30e-04 | 4.40e-04 | 1.67e-01 | 52.2 | 0.987 | 0.986 | 2.04e-02 |
| | ViT$_{\text{SLP}}$[4] | 3.56e-04 | - | 1.16e-01 | 34.3 | - | 0.90 | - |

[1] (Dao et al., 2023), [2](Lim et al., 2025b; Song & Ermon, 2020), [3](Davtyan et al., 2023), [4](Lim et al., 2025b); results reported from original paper trained on same dataset., [5](Lipman et al., 2023)

TempO. The orders of magnitude better next step prediction can be seen to blur significantly in rollout visualisations, provided for top next-step model FNO-2D in Appendix N

We select top performing comparisons for SWE and RD-2D, Table 3), where TempO maintains superior performance. In SWE, it achieves a 28.8% lower SpectralMSE and higher PSNR, indicating faithful amplitude, spectral content, and structural coherence with sharp boundaries preserved, see Appendix J for additional visualisations and ablated comparisons. Overall MSE is reduced by 30.8

In RD-2D, U-Net$_{\text{Affine-OT}}$ competes closely with TempO, benefiting from translation-equivariant convolutional layers that capture multi-scale dynamics and repeating local structures (Cohen & Welling, 2016). Both TempO and the U-Net have nearly matched PSNR, Pearon, and SSIM scores, with an improvement of 11% in SpectralMSE from the TempO. By contrast, the next best ViT regressor model is 95.6% drop in SpectralMSE, where attention might emphasize low-frequency global structures (Wang et al., 2022; Piao et al., 2024); see visual comparison in Appendix K.

The timeseries forecasting task, see Fig. 1, evaluates how well models capture the underlying PDE. Models follow the inference protocol used by (Davtyan et al., 2023; Li et al., 2021): a short lead-up of 9 initial frames is provided, with the baseline FNO using all 9; the sparsely conditioned flow-matching models use only the last two frames in the lead-up as conditioning and reference frames. The conditioning frame is then pinned and the temporal offset vector is incremented while the reference frame is set to timestep $t - 1$ to generate timestep $t$. TempO maintains Pearson correlation above 0.98 over 40 forecasted timesteps, indicating stable amplitude and phase tracking. The ViT regressor holds above 0.95 for 20 steps before degrading, while the flow matching baseline (Lipman et al., 2023) shows steady decline. This suggests TempO effectively mimics the dynamics without significant error accumulation. This is further demonstrated by visualisations of the vorticity field at key timesteps in Fig. 1 (right), where $t = 35$ most clearly shows TempO's faithful capture of turbulent eddies in comparison to the ViT regressor, which fails to predict the small vortical structure.

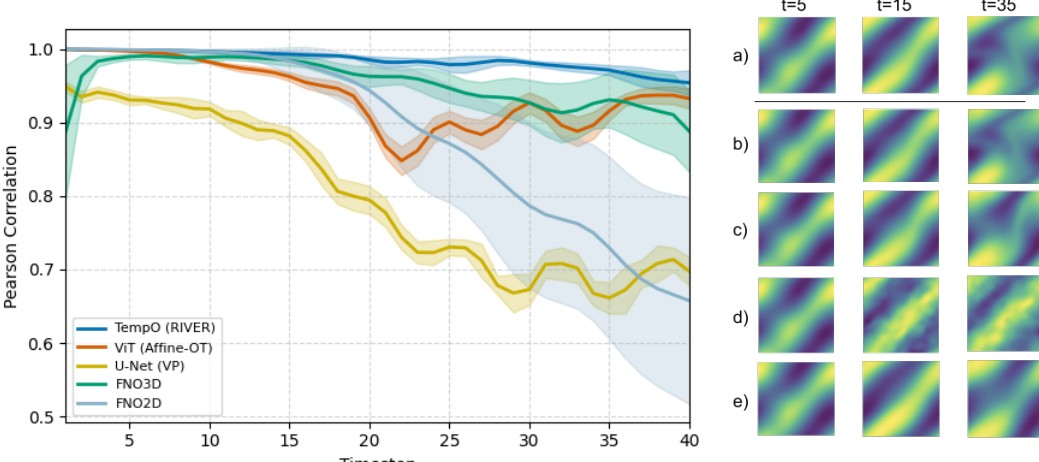

Figure 1: **Prediction performance comparison for NS-$\omega$.** *Left: Pearson correlation across forecasted timesteps.* Forty timesteps are predicted by TempO, ViT, U-Net, and top performing baseline FNO-3D averaged over 20 initial conditions on the withheld test set plotted as means and standard deviations. The Pearson correlation coefficient shows significant degradation for the U-Net, oscillatory behavior and degradation for the ViT, and consistently stable values above $0.95$ for TempO. The FNO-3D baseline exhibits a high variance in the first timesteps, attributed to the direct time convolution *Right: Predicted vorticity fields.* True data (a), TempO (b), ViT$_{\text{Affine-OT}}$ (c), U-Net$_{\text{VP-diff}}$, (d), and baseline FNO3D (e). At timesteps 5, 15, and 35 (c), (d), and (e) clearly diverge, with (d) regressing to a noisy diagonal and (e) losing detail, while TempO maintains excellent accuracy.

## 5.1 SPECTRAL ANALYSIS

The spectral analysis of TempO versus the top alternative ViT$_{\text{Affine-OT}}$ and the baseline U-Net$_{\text{Affine-OT}}$ (Lipman et al., 2023) in Fig. 2 examines the scale-resolved error via the energy per wavenumber $k$, or at the scale of $\frac{1}{k}$. This provides scale-resolved context to the SpectralMSE, which averages the MSE of the Fourier coefficients to a single metric. For NS-$\omega$, the first 8 modes which cumulatively make up 99% of the total energy, beyond which the modes have negligible contributions to overall flow dynamics, see Appendix H. TempO closely follows the true spectrum compared to both ViT$_{\text{Affine-OT}}$ and U-Net$_{\text{Affine-OT}}$, though all three methods diverge past $k = 8$. We observe from the inset of Fig. 2 that TempO exhibits a small residual which fluctuates about 0 whereas the ViT$_{\text{Affine-OT}}$ has a negative and increasing error: the ViT regressor tends to capture the lower wavenumbers well, but then underestimates the higher wavelengths notably after $k = 4$.

| Modes | SpectralMSE |
|-------|-------------|
| 1     | 8.57e-02    |
| 2     | 4.10e-02    |
| 4     | 3.98e-02    |
| 8     | 3.79e-02    |
| 16    | 3.74e-02    |

Figure 3: Ablation: Fourier mode cutoffs with TempO.

We observe also that the number of modes retained during the FFT of TempO in Fig. 3 follows the observation of a close spectral match up until $k = 8$, where the SpectralMSE sees the most improvement; however, from 8 modes to 16 modes, the performance appears to saturate. Fig. 3 demonstrates that up to 8 modes capture the essential dynamics, while the fundamental frequency alone is insufficient and likely under-represents necessary higher frequency components; adding more than 8 modes yields diminishing returns, matching the true spectral analysis; extended metrics support this trend in Appendix L. This empirical saturation beyond 8 modes is consistent with the theoretical expectation in Theorem 3.3, where FNOs are shown to achieve accuracy with asymptotically fewer parameters by leveraging only the most informative spectral modes.

To assess whether TempO's forecasting performance depends strongly on the spectral bandwidth of the latent FNO, we conduct an ablation in which the number of retained Fourier modes is varied (Appendix M). We find that the model is remarkably stable across a wide range of truncation levels: reducing from 16 to 8 or even 4 modes yields nearly identical MSE, spectral error, and long-horizon stability. This aligns with the spectral energy distribution observed in Fig. 2, where NS-$\omega$ exhibits

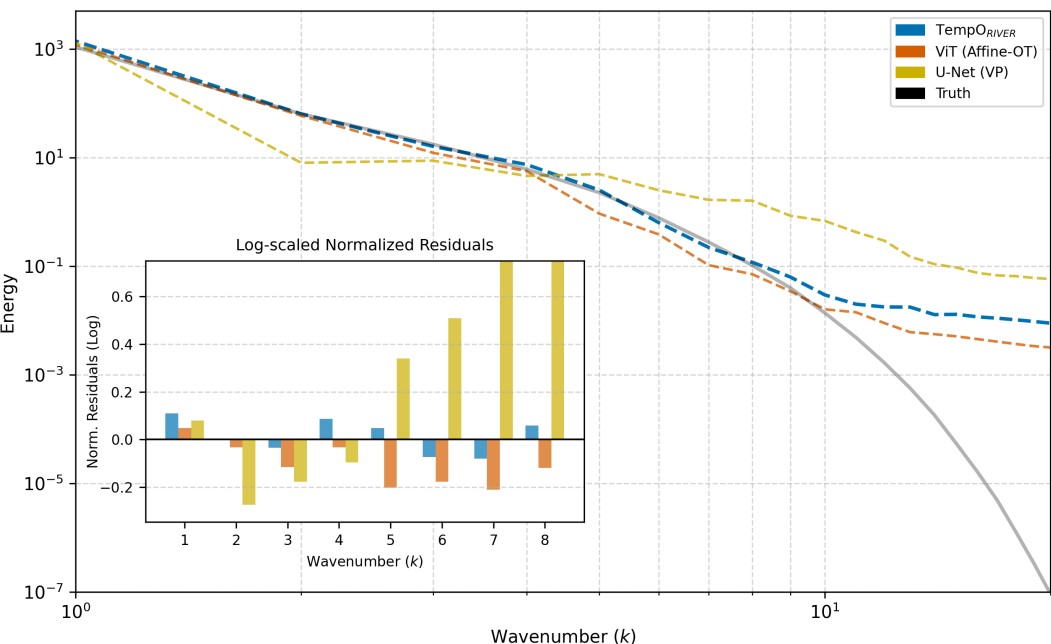

Figure 2: **Spectral graphs for NS-$\omega$.** Comparison of energy spectra for TempO, a ViT-based model, and the U-Net baseline (Lipman et al., 2023). The first eight Fourier modes capture 99% of the energy, with higher modes negligible. TempO aligns closely, while the ViT underestimates energy beyond $k = 4$. The inset bar plot shows TempO oscillating tightly around zero with small deviations, the ViT producing larger negative deviations, and the U-Net performing markedly worse.

weak high-frequency content. Only highly restrictive truncation, e.g., 2 or 1 mode, produces notice-able degradation, indicating that the autoencoder sufficiently captures the multiscale features and further confirming that TempO is robust to spectral compression in latent space.

## 5.2 EFFICIENCY

Finally, we also train the models over varying sequence lengths and measuring next-step prediction error (MSE) and 40-step forecast error (MSE/time), shown in Table 5. MSE is lowest for shorter sequences, as the model learns from fewer choices of indices for sparse conditioning during training. Conversely, MSE/time decreases with longer sequences, reflecting better long-horizon performance. Notably, TempO's MSE/time drops faster and plateaus lower than the ViT, indicating better data efficiency to extrapolate from the same available sequence length.

| Model | Params | FLOPs | Mem (MB) | NFEs |
|-------|--------|-------|----------|------|
| TempO | 0.49M | 208M | $\sim$50 | 560 |
| ViT | 3.39M | 10M | $\sim$80 | 942 |
| U-Net | 14.0M | 555M | $\sim$68 | 728 |

Table 4: Model Complexity and Efficiency: num-ber of function evaluationss (NFEs) are averaged from sampling performed for Table 2 for adaptive solver `dopri5` and tolerances of 1e-5.

TempO is the most lightweight model among the three choices of regressors, with $\tilde{7}$x fewer parame-ters than the ViT and $\tilde{28}$x fewer than the U-Net. In addition, it sees a significantly lower memory usage compared to the ViT where attention has higher de-mands and the U-Net where skip-connections hold onto additional memory.

| Method | Seq | MSE | MSE/time |
|--------|-----|-----|----------|
| TempO | 2 | 4.92e-02 | 2.70e-01 |
| | 5 | 4.75e-02 | 3.41e-01 |
| | 10 | 5.04e-02 | 4.94e-02 |
| | 15 | 5.61e-02 | 3.83e-02 |
| | 25 | 6.26e-02 | 4.22e-02 |
| ViT (Affine-OT) | 2 | 6.75e-02 | 2.71e-01 |
| | 5 | 5.43e-02 | 3.59e-01 |
| | 10 | 6.01e-02 | 1.49e-01 |
| | 15 | 6.70e-02 | 4.53e-02 |
| | 25 | 7.68e-02 | 8.56e-02 |

Table 5: Ablation: Performance comparison scaling with sequence length tested on NS-$\omega$.

While TempO has a moderate number of FLoating Point OPerations (FLOPs), landing between the ViT and U-Net, this may be offsetted by the NFEs seen during the ODE integration where TempO takes only 560 evaluations to meet the same tolerances. Beyond these empirical measures, TempO further benefits from its shared spatial Fourier layers. By folding the channel dimension and truncating higher modes, the spectral convolution scales as $O(N^2 \log N)$, in contrast to the naive $O(N^3 \log N)$ cost of a full 3D FFT. Also for reference, a ViT layer can scale as $O(N^4)$ in 2D 3.2, higher than the quasi-quadratic cost of the FNO.

## 6 LIMITATIONS

Flow matching models struggle with extreme data sparsity which can distort the distributions being learned, whereas hybrid models or models with explicitly defined conservations can fall back on injected physical knowledge. Additionally, similar to other generative models, adaptations, e.g. architectural modifications, would be necessary to extend the method towards a foundational model framework. Finally, while our stable and accurate 40-step forecasting represents the longer end time horizons, it remains an open question on how to forecast for much longer timeframes. Critical applications in science and engineering would require further study both experimentally and theoretically to establish statistically reliable forecasting.

## 7 CONCLUSIONS AND FURTHER WORK

In this work, we addressed the challenge of long-horizon PDE forecasting via our proposed method TempO. TempO consistently outperformed state-of-the-art baselines across three benchmark PDE datasets and achieves stable long-horizon 40 step forecasts with remarkable accuracy to the true trajectories as well as superior spectral fidelity. The modified time-conditioned FNO is parameter-efficient while improving the capture of both local and global spectral modes, resulting in improvements in both data- and compute- efficiency. Additionally, we establish that FNO can achieve an upper bound on approximation error that sampler-based architectures cannot reach without significantly more parameters, Corollary 3.3. These results highlight the importance of architectures that align with the continuous nature of PDE dynamics, enabling not only improved predictive accuracy but also physically consistent, long-horizon trajectories.

Consequently, TempO poses significant opportunity for further work in this field. Under typical real-world environments, PDE observations may come from irregularly sampled domains; since our method already demonstrates state-of-the-art generations using a simple autoencoder (AE) and the latent time-conditioned FNO which no longer relies on a regular grid as is a limitation of the original FNO (Li et al., 2021), one extension of our work is to then extend our method to real-world settings to forecast PDE over irregular domains and irregularly sampled domains. In addition, a detailed sensitivity analysis quantifying how TempO's effective Lipschitz constants and integration error evolve over long horizons would complement our empirical findings.

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

# A    PROOFS

*Proof of Theorem 3.1.* **Step 1: Spectral truncation.** By assumption the Fourier coefficients of $\mathcal{G}(u)$ satisfy

$$\left|\widehat{\mathcal{G}(u)}(k)\right| \leq C_\lambda (1 + |k|)^{-p}, \qquad \forall u \in \mathcal{U}, \ \ k \in \mathbb{Z}^d.$$

If we keep only the modes $|k| \leq K$ and set

$$\mathcal{G}_K(u)(x) := \sum_{|k| \leq K} \widehat{\mathcal{G}(u)}(k) e^{ik \cdot x},$$

then the error lives in the high modes:

$$\|\mathcal{G}(u) - \mathcal{G}_K(u)\|_{H^{s'}}^2 = \sum_{|k| > K} (1 + |k|^2)^{s'} \left|\widehat{\mathcal{G}(u)}(k)\right|^2.$$

Using the decay bound gives

$$\|\mathcal{G}(u) - \mathcal{G}_K(u)\|_{H^{s'}}^2 \leq C_\lambda^2 \sum_{|k| > K} (1 + |k|)^{2(s' - p)}.$$

A standard counting argument (comparing the lattice sum with a radial integral) shows this tail is $\lesssim K^{-2\alpha}$, with

$$\alpha := p - s' - \frac{d}{2} > 0.$$

This is exactly the pseudo-spectral tail estimate also used in (Kovachki et al., 2021, Thm. 40). Hence choosing

$$K \asymp \varepsilon^{-1/\alpha}$$

ensures $\|\mathcal{G} - \mathcal{G}_K\|_{H^{s'}} \leq \varepsilon/2$.

**Step 2: Reduction to a finite-dimensional map.** The truncated operator $\mathcal{G}_K$ is determined by finitely many Fourier coefficients $\{\widehat{\mathcal{G}(u)}(k)\}_{|k| \leq K}$, with output dimension $m_{\text{out}} \asymp K^d$. To apply a neural network, we also restrict the input to finitely many low modes. By compactness of $\mathcal{U} \subset H^s$ and continuity of the projection $P_M$, there exists $M$ such that

$$\|\mathcal{G}_K(u) - \mathcal{G}_K(P_M u)\|_{H^{s'}} \leq \varepsilon/6 \qquad \forall u \in \mathcal{U}.$$

This is the same finite-dimensional reduction used in the universal approximation argument of (Kovachki et al., 2021, Thm. 15). Thus it suffices to approximate the finite-dimensional continuous map

$$F : (\widehat{u}(k))_{|k| \leq M} \longmapsto (\widehat{\mathcal{G}(u)}(k))_{|k| \leq K},$$

between compact subsets of Euclidean spaces.

**Step 3: Approximation of the finite map.** Classical universal approximation theorems (and the constructive $\Psi$–FNO realization in (Kovachki et al., 2021, Def. 11, Thm. 15)) ensure that for any desired accuracy $\delta > 0$, one can build a neural network (or FNO block) approximating $F$ uniformly to error $\delta$ on each retained coefficient. To control the $H^{s'}$–norm it suffices to achieve coefficient accuracy

$$\delta \lesssim \frac{\varepsilon}{K^{s' + d/2}}.$$

This choice ensures $\|P_K \mathcal{G}(u) - \widetilde{\mathcal{G}}_\theta(u)\|_{H^{s'}} \leq \varepsilon/3$. Constructive approximation bounds then give a parameter count

$$P \lesssim K^d \cdot \text{polylog}(1/\varepsilon),$$

where the extra logarithmic factor reflects standard overheads in coefficient quantization and network approximation (Kovachki et al., 2021, Remark 22).

**Step 4: Assemble errors and conclude.** Adding the contributions: - spectral truncation error $\leq \varepsilon/2$ (Step 1), - input-projection error $\leq \varepsilon/6$ (Step 2), - finite-map approximation error $\leq \varepsilon/3$ (Step 3),

we obtain

$$\sup_{u \in \mathcal{U}} \|\mathcal{G}(u) - \mathcal{G}_\theta(u)\|_{H^{s'}} \leq \varepsilon.$$

Substituting $K \asymp \varepsilon^{-1/\alpha}$ into the parameter bound gives

$$P_{\mathrm{FNO}}(\varepsilon) \lesssim \varepsilon^{-d/\alpha},$$

up to the mild logarithmic factors discussed above. □

*Proof of Proposition 3.2. Step 1: Finite-dimensional subspace and sampling.* Consider the $K$-mode Fourier subspace

$$V_K := \mathrm{span}\{e^{ik \cdot x} : |k| \leq K\} \subset L^2(\mathbb{T}^d), \qquad \dim V_K =: D_K \asymp K^d.$$

Any sampler-based learner observes an input $u \in V_K$ only through $n$ fixed points $(u(x_1), \dots, u(x_n))$. This defines a linear map

$$S : V_K \to \mathbb{C}^n, \quad S(u) = (u(x_1), \dots, u(x_n)).$$

*Step 2: Nyquist / injectivity argument.* To reconstruct all Fourier modes up to radius $K$, the sampling map $S$ must be injective on $V_K$. In matrix terms, $S$ is represented by an $n \times D_K$ Vandermonde-like matrix. To have full rank $D_K$, we require

$$n \geq D_K \asymp K^d.$$

If $n < D_K$, there exists a nonzero $u \in V_K$ vanishing on all sample points, so the learner cannot distinguish it from zero. This is the standard Nyquist/dimension-counting requirement: at least as many samples as degrees of freedom.

*Step 3: Parameter lower bound.* After sampling, the learner applies a parametric map $M : \mathbb{C}^n \to \mathbb{C}^m$ (e.g., a neural network) to produce either output samples or coefficients. To implement arbitrary linear transformations on the $D_K$ retained modes (e.g., arbitrary Fourier multipliers), the parametric map must have at least $D_K$ free parameters. For fully general dense linear maps (no structural constraints), one needs

$$P \gtrsim D_K^2 \asymp K^{2d}.$$

*Step 4: Conversion to accuracy $\varepsilon$.* From the FNO upper bound analysis, achieving accuracy $\varepsilon$ requires

$$K \asymp \varepsilon^{-1/\alpha}, \qquad \alpha = p - s' - d/2 > 0.$$

Substituting this into the previous bounds gives the scaling

$$n \gtrsim \varepsilon^{-d/\alpha}, \qquad P_{\mathrm{sampler}}(\varepsilon) \gtrsim \varepsilon^{-\beta d/\alpha},$$

with $\beta = 1$ for minimal mode-wise maps and $\beta = 2$ for fully dense maps.

*Step 5: Conclusion.* Hence any sampler-based architecture that must reconstruct all modes up to radius $K$ requires asymptotically more parameters than an FNO whenever $\beta > 1$, justifying the lower bound in the proposition. □

## B   FLOW MATCHING BACKGROUND

**Flow matching**   The core idea of flow matching is to learn a time-dependent velocity field, $v_\theta(z, t)$, which defines an ODE in the latent space:

$$\frac{dz(t)}{dt} = v_\theta(z(t), t), \quad z(0) \sim \pi_0, \tag{2}$$

where $\pi_0$ is a simple reference distribution (e.g., Gaussian). Integrating this ODE transports samples to the latent data distribution $\pi_1$, such that $z(1) \sim \pi_1$ and $p_1(z) \approx f_\phi \# \mathcal{D}_{\mathrm{data}}$, where $f_\phi \# \mu$ denotes the pushforward measure of a distribution $\mu$ under $f_\phi$, i.e., $(f_\phi \# \mu)(A) = \mu(f_\phi^{-1}(A))$ for

measurable sets $A$. The corresponding time-dependent probability density, $p_t(z)$, evolves according to the continuity equation:

$$\frac{\partial p_t(z)}{\partial t} + \nabla_z \cdot \big(p_t(z)\, v_\theta(z,t)\big) = 0. \tag{3}$$

In practice, the target velocity field $u(t, z)$ and the full marginal density $p_t(z)$ are generally unknown and intractable. Flow matching sidesteps this issue by directly supervising the model to match the instantaneous vector field along interpolating paths between the reference $\pi_0$ and the target $\pi_1$, allowing for deterministic, efficient sampling. Different choices of paths lead to different training dynamics and inductive biases, as they implicitly define the target velocity field $u(t, z)$ that the model regresses against.

Integrating this ODE from $t = 0$ to $t = 1$ transports the reference distribution $\pi_0$ to the latent data distribution $\pi_1$, so that $z(1) \sim \pi_1$ and $p_1(z) \approx f_\phi \# \mathcal{D}_{\text{data}}$.

**Latent Flow Matching.** We now instantiate the general flow matching framework in the latent space. Let $z_\tau = f_\phi(x_\tau)$ for $\tau = 1, \dots, m$, where $f_\phi$ is a pretrained encoder mapping from the data space to the lower-dimensional latent space. Our objective is to approximate the ground-truth latent distribution $q(z_\tau \mid x_1, \dots, x_{\tau-1})$ by a parametric distribution $p(z_\tau \mid z_{\tau-1})$, which can later be decoded back to the data space via $x_\tau = g_\psi(z_\tau)$ using a decoder $g_\psi$.

The latent dynamics can be expressed by the ODE:

$$\dot{z}_t = u_t(z_t), \tag{4}$$

where $u_t$ denotes the (true) time-dependent velocity field. Learning these dynamics amounts to approximating $u_t$ with a neural parameterization. Following the flow matching framework, we introduce a model velocity field $v_\theta : [0, 1] \times \mathbb{R}^Z \to \mathbb{R}^Z$ and consider the ODE

$$\dot{\phi}_t(z) = v_\theta(\phi_t(z), t), \quad \phi_0(z) = z, \tag{5}$$

which defines a time-dependent diffeomorphism $\phi_t$ pushing forward an initial reference distribution $p_0$ (often chosen as $\mathcal{N}(0, I)$) to a target distribution $p_1 \approx q$ along the density path $p_t$:

$$p_t = (\phi_t)_\# p_0, \tag{6}$$

where $(\cdot)_\#$ denotes the pushforward. In other words, the goal of flow matching is to learn a deterministic coupling between $p_0$ and $q$ by training $v_\theta$ so that the solution satisfies $z_0 \sim p_0$ and $z_1 \sim q$.

Given a probability path $p_t$ and its associated velocity field $u_t$, flow matching reduces to a least-squares regression problem:

$$\mathcal{L}_{\text{FM}}(\theta) = \mathbb{E}_{t \sim U[0,1],\, z \sim p_t}\, \omega(t) \, \|v_\theta(z, t) - u_t(z)\|_2^2, \tag{7}$$

where $\omega(t) > 0$ is a weighting function, often taken as $\omega(t) = 1$ (Lipman et al., 2022). This formulation ensures that the learned velocity field aligns with the target field $u_t$ at all times, thereby generating the desired marginal probability path.

## C   FOURIER NEURAL OPERATOR BACKGROUND

An FNO is designed to learn a mapping between function spaces, rather than between finite-dimensional vectors. Consider a function $u : \mathbb{R}^d \to \mathbb{R}^c$ representing data, for example in $\mathbb{R}^X$, with samples $x \in \mathbb{R}^X$. Then, an FNO parameterizes an operator as

$$\mathcal{G}_\theta : u \mapsto \tilde{u}, \quad \tilde{u} : \mathcal{D} \to \mathbb{R}^{c_{\text{out}}},$$

that maps $u$ to an output function $\tilde{u}$ (e.g., a solution field of a PDE or a transformed spatial signal).

This mapping is implemented via iterative Fourier layers which perform spectral transformations of the input:

$$\hat{u}(k) = \mathcal{F}[u](k), \quad \hat{\tilde{u}}(k) = R_\theta(k) \cdot \hat{u}(k), \tag{8}$$

followed by an inverse Fourier transform back to the spatial domain:

$$\tilde{u}(x) = \mathcal{F}^{-1}[\hat{\tilde{u}}](x), \tag{9}$$

with $R_\theta(k)$ being learnable Fourier-mode weights and $\mathcal{F}$ denoting the Fourier transform. This spectral representation allows the FNO to efficiently capture long-range dependencies and global correlations in the data.

## D  AUTOENCODER DETAILS

Residual blocks throughout the architecture consist of two $3 \times 3$ convolutions with ReLU activation and group normalization (8 groups) in between, with the input added back to the output. Attention blocks are implemented using PyTorch's `nn.MultiheadAttention`, with embeddings reshaped from $[B, C, H, W]$ to $[B, HW, C]$.

The autoencoder is initialised with a depth of $d = 2$ resulting in a factor $2^d = 4$ compression for all datasets.

## E  MODEL HYPERPARAMETERS

We initialised the probability paths with the following hyperparameters. RIVER was defined with variance parameters $\sigma = 0.1$ and $\sigma_{\min} = 10^{-7}$. SLP used $\sigma = 0.1$ and $\sigma_{\min} = 0.01$. We further considered the VE-diff path with $\sigma_{\min} = 0.01$ and $\sigma_{\max} = 0.1$ and the VP-diff path initialized with $\beta_{\min} = 0.1$ and $\beta_{\max} = 20.0$ per (Lim et al., 2025b).

We provide details for the vector field regressors' width and depth hyperparameters as per Table 6.

| Model | Parameter | Value |
|-------|-----------|-------|
| TempO | $n_{\mathrm{modes}}$ | 20 |
|       | Hidden channels | 64 |
|       | Projection channels | 64 |
|       | Depth | 4 |
| U-Net | Hidden channels | 64 |
|       | Attention resolutions | (1, 2, 2) |
|       | Channel multiplier | (1, 2, 4) |
|       | Depth | 3 |
| ViT   | Hidden channels | 256 |
|       | Depth | 4 |
|       | Mid-depth | 5 |
|       | Output normalization | LayerNorm |

Table 6: Descriptions of hyperparameters across TempO, U-Net, and ViT architectures.

## F  TRAINING AND INFERENCE SETUP FOR BASELINE MODELS

We evaluate three classes of baselines: Fourier Neural Operators (FNO) (Li et al., 2021), Wavelet Neural Operators (WNO) (Tripura & Chakraborty, 2022), and a standard U-Net backbone.

**2D Autoregressive Models (FNO-2D, WNO-2D, U-Net).**  The 2D variants of FNO, WNO, and U-Net operate purely on spatial fields $\omega(\cdot, t) \in \mathbb{R}^{64 \times 64}$ and treat time autoregressively. Each model is provided a dense block of $n$ input frames. These frames are concatenated and mapped to the next-time-step prediction $\hat{\omega}(\cdot, t + 1)$. During training, models minimize an $\ell_2$ regression loss on the next-step vorticity. During inference, the prediction is appended to the input sequence, and the model is rolled forward autoregressively for the desired horizon, replacing the oldest frame at each step. This *2D+RNN* structure allows propagation to arbitrarily long forecast windows using a fixed temporal stride.

**3D Convolutional Models (FNO-3D, WNO-3D).**  The 3D variants of FNO and WNO treat time as an additional convolutional dimension and directly process space–time blocks $(x, y, t)$ as 3D volumes. Following Li et al. (2021), the model receives the dense history of the first $n$ timesteps and performs a 3D convolutional operator mapping

$$\mathbb{R}^{n \times x \times y} \longrightarrow \mathbb{R}^{T_{\mathrm{pred}} \times x \times y},$$

producing the entire future trajectory segment $T$ at once. Unlike the 2D autoregressive setting, the 3D operator does *not* iterate forward in time: it learns a direct operator from the initial block to the entire forecast window. This makes the method better conditioned and more expressive at the cost of requiring a fixed temporal window during training (Li et al., 2021).

Both 2D and 3D WNOs follow the same temporal training structure as their FNO counterparts. The only architectural change is the replacement of Fourier transforms with multi-resolution wavelet transforms, but the data flow (input–output tensor shapes, roll-out strategy, and supervision) is identical.

**Training Details.**   Models were trained using the Adam optimiser with weight decay of 0.0001. FNO and WNO learning rate of 0.001 except the U-Net, which was trained with a learning rate of 0.0001.

| Model | Dim | Layers | Width | Modes / Wavelet | Level | Input Ch. |
|---|---|---|---|---|---|---|
| FNO-3D | 3D | 4 | 64 | $(20, 20, 16)$ (Fourier) | – | 1 |
| FNO-2D | 2D | 4 | 64 | $(20, 20)$ (Fourier) | – | $T_{\text{in}}$ |
| UNet-2D | 2D | 4 (down/up) | 128 base | – | – | $T_{\text{in}}$ |
| WNO-3D | 3D | 4 | 40 | db6 (wavelet) | 2 | $(T_{\text{in}} + 3)$ |
| WNO-2D | 2D | 4 | 64 | db6 (wavelet) | 2 | $(T_{\text{in}} + 2)$ |

Table 7: Summary of hyperparameters for Fourier and Wavelet neural operators and the UNet baseline. Channels refers to hidden width. Modes applies only to FNO; wavelet type/level applies only to WNO.

**Inference Protocol.**   For comparability with prior operator-learning works, we evaluate autoregressive models by providing the first $n = 10$ frames as context and generating the next 40 timesteps using a single-step temporal stride. For 3D models, the first $n$ frames are mapped directly to a 40-step output block without iterative rollout. This setup follows the conventions established in (Li et al., 2021; Tripura & Chakraborty, 2022).

# G   DATASET DETAILS

Table 8: Dataset sizes and trajectory lengths used in evaluation.

| Dataset | # Trajectories | Timeseries Length |
|---|---|---|
| SWE | 1000 | 100 |
| RD-2D | 1000 | 100 |
| NS-$\omega$ | 5000 | 50 |

*Shallow water equation (SWE)*

The SWEs are derived from the compressible Navier–Stokes equations and model free-surface flow problems in 2D. The system of hyperbolic PDEs is given by:

$$\partial_t h + \partial_x(hu) + \partial_y(hv) = 0, \tag{10}$$

$$\partial_t(hu) + \partial_x\left(u^2 h + \frac{1}{2}g_r h^2\right) + \partial_y(uvh) = -g_r h \, \partial_x b, \tag{11}$$

$$\partial_t(hv) + \partial_y\left(v^2 h + \frac{1}{2}g_r h^2\right) + \partial_x(uvh) = -g_r h \, \partial_y b, \tag{12}$$

where $u, v$ are the horizontal and vertical velocities, $h$ is the water height, $b$ represents spatially varying bathymetry, and $g_r$ is gravitational acceleration. The quantities $hu$ and $hv$ correspond to directional momentum components.

The dataset ( (Takamoto et al., 2022)) simulates a 2D radial dam break scenario on a square domain $\Omega = [-2.5, 2.5]^2$. The initial water height is a circular bump in the center of the domain:

$$h(t = 0, x, y) = \begin{cases} 2.0, & \text{if } r < r_0, \\ 1.0, & \text{if } r \geq r_0, \end{cases} \quad r = \sqrt{x^2 + y^2}, \quad r_0 \sim \mathcal{U}(0.3, 0.7).$$

*2D reaction diffusion (RD-2D)*

The RD-2D dataset models two non-linearly coupled variables: the activator $u = u(t, x, y)$ and the inhibitor $v = v(t, x, y)$. The system of PDEs is:

$$\partial_t u = D_u \, \partial_{xx} u + D_u \, \partial_{yy} u + R_u(u, v), \tag{13}$$
$$\partial_t v = D_v \, \partial_{xx} v + D_v \, \partial_{yy} v + R_v(u, v), \tag{14}$$

where $D_u$ and $D_v$ are diffusion coefficients, and $R_u(u, v)$, $R_v(u, v)$ are the reaction functions. Specifically, the FitzHugh–Nagumo model defines the reactions as:

$$R_u(u, v) = u - u^3 - k - v, \tag{15}$$
$$R_v(u, v) = u - v, \tag{16}$$

with $k = 5 \times 10^{-3}$, $D_u = 1 \times 10^{-3}$, and $D_v = 5 \times 10^{-3}$.

The dataset ( (Takamoto et al., 2022)) uses a simulation domain $x, y \in (-1, 1)$ and $t \in (0, 5]$ with initial condition set as standard normal random noise: $u(0, x, y) \sim \mathcal{N}(0, 1.0)$.

*2D incompressible Navier-Stokes vorticity (NS-$\omega$)*

The NS-$\omega$ ( (Li et al., 2021)) models 2D incompressible fluid flow on the unit torus. The system of equations is:

$$\partial_t w(x, t) + u(x, t) \cdot \nabla w(x, t) = \nu \, \Delta w(x, t) + f(x), \quad x \in (0, 1)^2, \, t \in (0, T], \tag{17}$$
$$\nabla \cdot u(x, t) = 0, \tag{18}$$
$$w(x, 0) = w_0(x), \tag{19}$$

where $w(x, t)$ is the vorticity, $u(x, t)$ is the velocity field, $\nu$ is viscosity, and $f(x)$ is a fixed forcing term:

$$f(x) = 0.1 \Big( \sin(2\pi(x_1 + x_2)) + \cos(2\pi(x_1 + x_2)) \Big).$$

The initial condition is sampled from a Gaussian measure:

$$w_0 \sim \mu, \quad \mu = \mathcal{N}\Big(0, \, \left(-\Delta + 49I\right)^{-2.5} 7^{3/2}\Big),$$

with periodic boundary conditions.

# H    SPECTRAL ANALYSIS OF GROUND TRUTH NS-$\omega$

Fig. 4 shows how the quality of spectral truncations of the true Navier–Stokes vorticity field depends on the cutoff wavenumber $k_{\text{cut}}$. Given the full Fourier spectrum $\hat{\omega}(k_x, k_y)$, we apply a mask that retains only modes with $|k_x| + |k_y| \leq k_{\text{cut}}$, reconstruct the signal by inverse FFT, and compute three quantities as functions of $k_{\text{cut}}$:

1. Reconstruction MSE: the mean squared error between the original and truncated fields in physical space.
2. Spectral MSE: the mean squared error in Fourier space, quantifying lost spectral content.

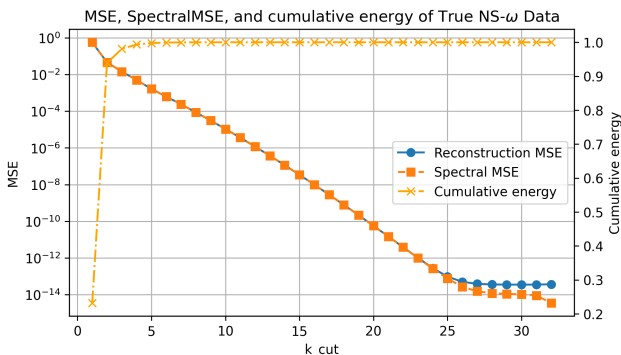

Figure 4: **Spectral Analysis of True Vorticity**: Reconstruction MSE, spectral MSE, and cumulative enstrophy fraction of true Navier–Stokes vorticity data as functions of cutoff wavenumber $k_{\mathrm{cut}}$.

3. Cumulative energy fraction: the fraction of total energy $\sum |\hat{\omega}|^2$ retained by the truncated spectrum.

As $k_{\mathrm{cut}}$ increases, both reconstruction and spectral errors decrease, while the retained energy approaches unity.

# I EXTENDED RESULTS FOR NAVIER–STOKES VORTICITY

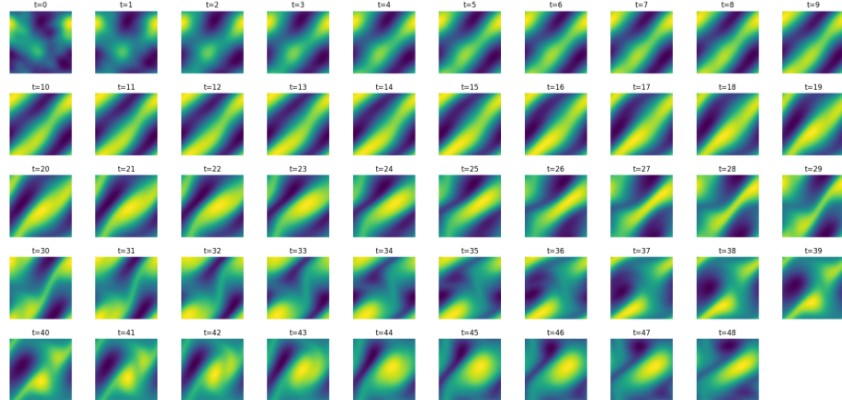

Figure 5: **Navier–Stokes Vorticity (Original).** Ground-truth timeseries across 40 timesteps.

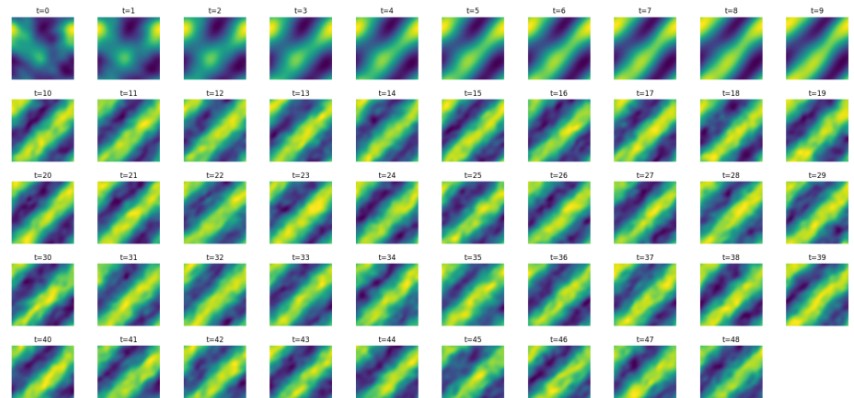

Figure 6: **Navier–Stokes Vorticity (U-Net).** Forecasted timeseries across 40 timesteps.

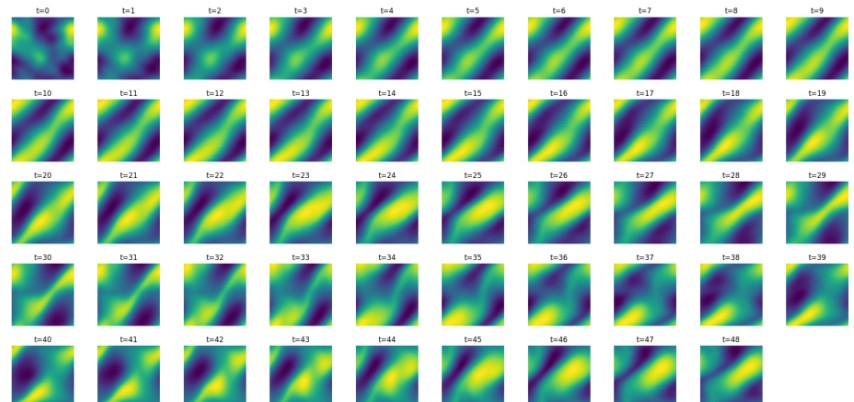

Figure 7: **Navier–Stokes Vorticity (ViT).** Forecasted timeseries across 40 timesteps.

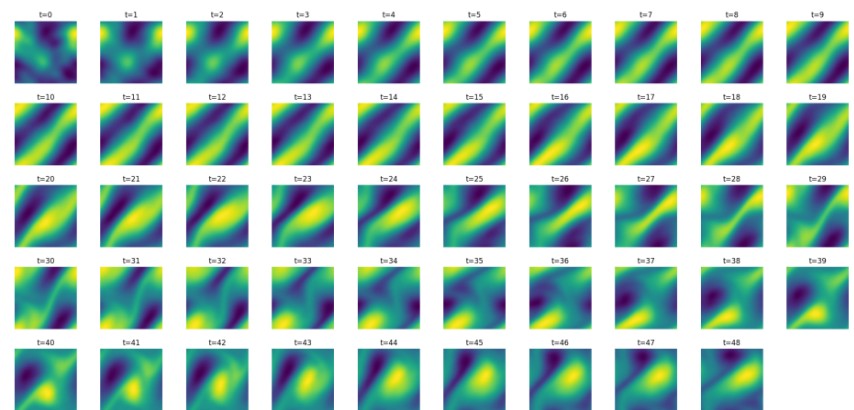

Figure 8: **Navier–Stokes Vorticity (TempO).** Forecasted timeseries across 40 timesteps.

## J    EXTENDED RESULTS FOR SHALLOW WATER EQUATION

| Regressor | Path | MSE ↓ | SpectralMSE ↓ | RFNE ↓ | PSNR ↑ | Pearson ↑ | SSIM ↑ |
|---|---|---|---|---|---|---|---|
| TempO | Affine-OT | **6.64e-05** | **5.65e-05** | **7.64e-03** | **46.5** | **0.998** | **0.997** |
| | RIVER | 4.04e-04 | 2.33e-04 | 1.89e-02 | 38.7 | 0.989 | 0.976 |
| | VE-diff | 9.37e-04 | 8.22e-04 | 2.89e-02 | 35.2 | 0.994 | 0.977 |
| | VP-diff | 4.41e-03 | 2.51e-03 | 4.31e-02 | 28.3 | 0.872 | 0.857 |
| ViT | Affine-OT | 9.59e-05 | 7.93e-05 | 9.06e-03 | 44.9 | 0.997 | 0.995 |
| | VP-diff | 1.30e-04 | 8.81e-05 | 1.05e-02 | 43.6 | 0.996 | 0.993 |
| | RIVER | 2.99e-04 | 1.67e-04 | 1.63e-02 | 40.0 | 0.992 | 0.981 |
| | SLP[1] | 6.60e-04 | - | 1.28e-01 | 36.1 | - | 0.93 |
| | VE-diff | 1.28e-03 | 1.01e-03 | 3.38e-02 | 33.7 | 0.985 | 0.960 |
| U-Net | VP-diff | 1.37e-02 | 8.26e-03 | 1.10e-01 | 23.4 | 0.546 | 0.627 |
| | RIVER | 1.61e-02 | 1.00e-02 | 1.20e-01 | 22.7 | 0.437 | 0.610 |
| | Affine-OT | 1.68e-02 | 1.01e-02 | 1.22e-01 | 22.5 | 0.435 | 0.593 |

Table 9: Comparison of TempO, U-Net, and ViT models under different probability paths for the SWE. The best value for each metric is highlighted in bold.

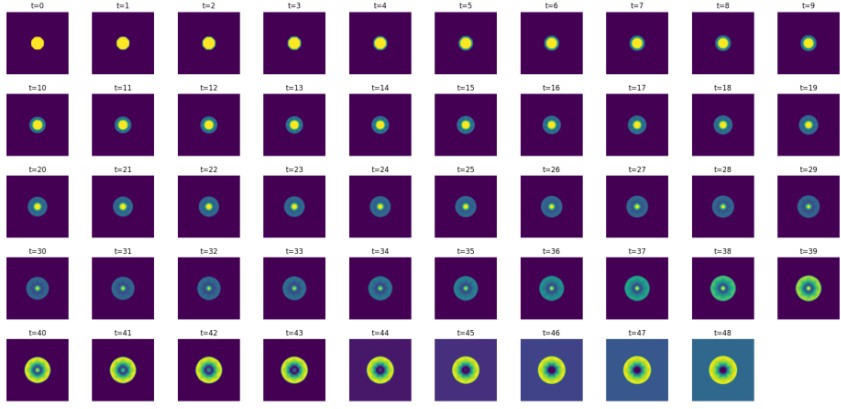

Figure 9: **SWE (Original).** Ground-truth timeseries across 40 timesteps.

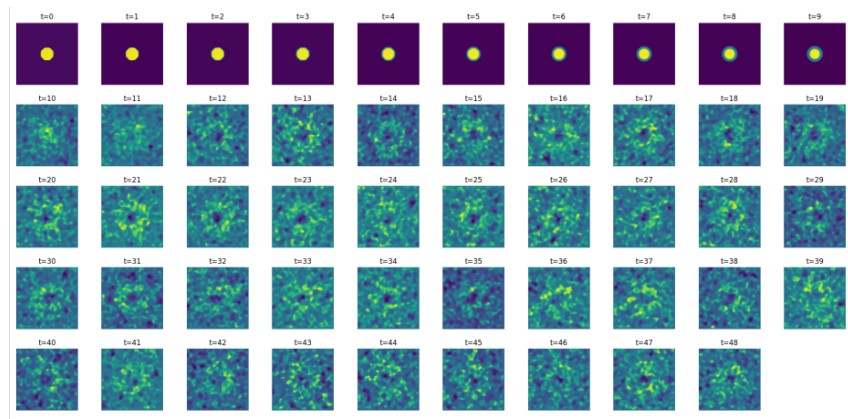

Figure 10: **SWE (U-Net).** Forecasted timeseries across 40 timesteps.

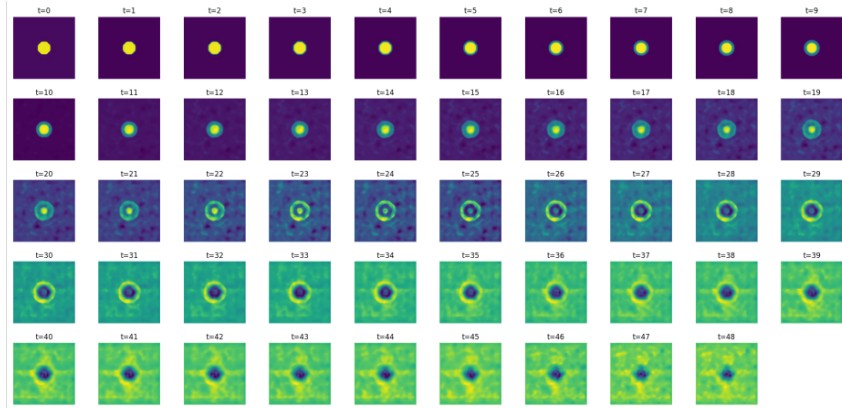

Figure 11: **SWE (ViT).** Forecasted timeseries across 40 timesteps.

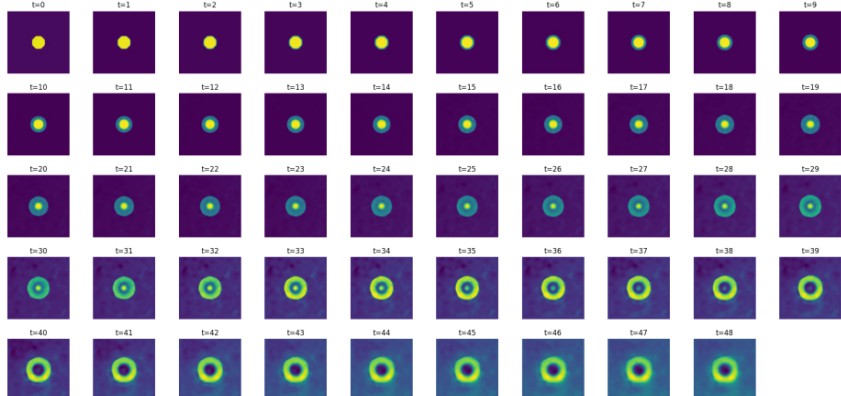

Figure 12: **SWE (TempO).** Forecasted timeseries across 40 timesteps.

## K EXTENDED RESULTS FOR 2D REACTION DIFFUSION

| Regressor | Path | MSE ↓ | SpectralMSE ↓ | RFNE ↓ | PSNR ↑ | Pearson ↑ | SSIM ↑ |
|---|---|---|---|---|---|---|---|
| TempO | Affine-OT | **2.76e-05** | **2.18e-05** | **3.29e-02** | **65.7** | **1.000** | **0.999** |
| | RIVER | 9.36e-04 | 5.47e-04 | 2.08e-01 | 50.4 | 0.975 | 0.978 |
| | VE-diff | 1.58e-03 | 1.38e-03 | 2.70e-01 | 48.2 | 0.990 | 0.977 |
| | VP-diff | 1.24e-02 | 1.01e-02 | 4.95e-01 | 39.2 | 0.714 | 0.862 |
| ViT | Affine-OT | 6.30e-04 | 4.40e-04 | 1.67e-01 | 52.2 | 0.987 | 0.986 |
| | SLP[2] | 3.56e-04 | - | 1.16e-01 | 34.3 | - | 0.90 |
| | RIVER | 1.00e-03 | 5.89e-04 | 2.16e-01 | 50.1 | 0.973 | 0.977 |
| | VE-diff | 3.54e-03 | 2.23e-03 | 4.06e-01 | 44.7 | 0.915 | 0.946 |
| U-Net | Affine-OT | 3.09e-05 | 2.45e-05 | 3.57e-02 | 65.2 | 0.999 | **0.999** |
| | RIVER | 1.02e-03 | 5.49e-04 | 2.17e-01 | 50.1 | 0.972 | 0.976 |
| | VE-diff | 9.03e-03 | 6.07e-03 | 6.42e-01 | 40.6 | 0.820 | 0.860 |
| | VP-diff | 2.09e-02 | 1.66e-02 | 6.81e-01 | 37.0 | 0.574 | 0.792 |

Table 10: Comparison of TempO, U-Net, and ViT models under different probability paths for the RD-2D. The best value for each metric is highlighted in bold.

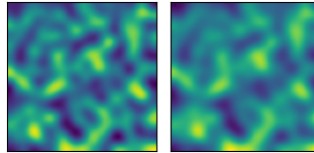

Figure 13: **Reaction Diffusion (Original).** Ground-truth end sample, from initial conditions of randomly sampled noise.

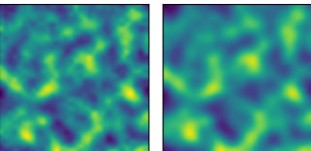

Figure 14: **Reaction Diffusion (U-Net).** Forecasted end sample, from initial conditions of randomly sampled noise.

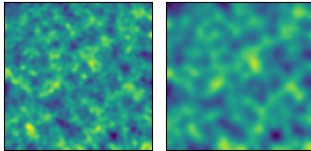

Figure 15: **Reaction Diffusion (ViT).** Forecasted end sample, from initial conditions of randomly sampled noise.

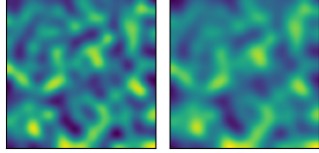

Figure 16: **Reaction Diffusion (TempO).** Forecasted end sample, from initial conditions of randomly sampled noise.

## L  EXTENDED ABLATION RESULTS

Table 11: Ablation over different training sequence lengths on the NS-$\omega$ dataset. TempO and the top performing alternative are trained while varying sequence lengths and evaluated on 10 timesteps to predict the next step.

| Method | Seq. Len. | MSE | DensityMSE | SpectralMSE | RFNE | PSNR | Pearson | SSIM | NFE |
|---|---|---|---|---|---|---|---|---|---|
| TempO | 3 | 4.924e-02 | 7.685e-05 | 3.531e-02 | 2.328e-01 | 25.769 | 0.973 | 0.803 | 74 |
| | 6 | 4.753e-02 | 1.133e-04 | 3.394e-02 | 2.276e-01 | 25.923 | 0.974 | 0.800 | 608 |
| | 11 | 5.036e-02 | 1.055e-04 | 3.620e-02 | 2.352e-01 | 25.672 | 0.972 | 0.800 | 842 |
| | 16 | 5.607e-02 | 1.282e-04 | 3.821e-02 | 2.497e-01 | 25.205 | 0.969 | 0.786 | 938 |
| | 26 | 6.255e-02 | 7.487e-05 | 3.726e-02 | 2.541e-01 | 24.730 | 0.968 | 0.765 | 1070 |
| ViT (Affine-OT) | 3 | 6.748e-02 | 1.414e-04 | 4.652e-02 | 2.678e-01 | 24.401 | 0.963 | 0.766 | 116 |
| | 6 | 5.434e-02 | 1.239e-04 | 3.727e-02 | 2.416e-01 | 25.341 | 0.970 | 0.783 | 1766 |
| | 11 | 6.014e-02 | 1.376e-04 | 4.067e-02 | 2.546e-01 | 24.901 | 0.967 | 0.777 | 1712 |
| | 16 | 6.701e-02 | 1.093e-04 | 4.428e-02 | 2.680e-01 | 24.431 | 0.963 | 0.764 | 1622 |
| | 26 | 7.682e-02 | 8.104e-05 | 4.468e-02 | 2.778e-01 | 23.838 | 0.960 | 0.741 | 1100 |

Table 12: Ablation of the TempO model on the NS-$\omega$ dataset by varying the number of modes. Models are trained with different numbers of Fourier modes and evaluated on 10 timesteps to predict the next step.

| Modes | MSE | DensityMSE | SpectralMSE | RFNE | PSNR | Pearson | SSIM | NFE |
|---|---|---|---|---|---|---|---|---|
| 1 | 1.409e-01 | 1.075e-04 | 8.566e-02 | 3.947e-01 | 21.204 | 0.921 | 0.588 | 5798 |
| 2 | 6.103e-02 | 8.928e-05 | 4.096e-02 | 2.596e-01 | 24.837 | 0.966 | 0.765 | 1688 |
| 4 | 5.789e-02 | 8.361e-05 | 3.978e-02 | 2.538e-01 | 25.066 | 0.968 | 0.776 | 1058 |
| 8 | 5.528e-02 | 8.498e-05 | 3.788e-02 | 2.481e-01 | 25.267 | 0.969 | 0.788 | 800 |
| 16 | 5.471e-02 | 8.757e-05 | 3.742e-02 | 2.467e-01 | 25.312 | 0.970 | 0.787 | 884 |

## M  ABLATION ON FOURIER-MODE TRUNCATION

**Setup**  To assess the sensitivity of TempO to the number of retained Fourier modes, we perform an ablation in which the spectral truncation level of the underlying FNO blocks is varied while keeping all other architectural and training settings fixed. Specifically, we evaluate truncation levels of

$$m \in \{16,\ 8,\ 4,\ 2,\ 1\},$$

where $m$ denotes the number of Fourier modes kept per spatial dimension. The case $m = 1$ represents the most extreme truncation and serves as a lower-bound sanity check for TempO under highly restricted spectral capacity.

**Method**  All models are trained with identical optimisation hyperparameters and dataset splits on the NS-$\omega$ dataset. For each truncation level, we measure ..., following the evaluation protocol used in the main experiments.

**Results**  Table 13 reports the aggregated metrics. We observe that performance remains stable for moderate truncations ($m = 8$ and $m = 16$), with only mild degradation for $m = 4$. As expected, substantial deterioration appears only when the spectral capacity is collapsed to $m = 2$ or $m = 1$, reflecting the loss of essential mid-frequency components.

**Discussion**  The study shows that TempO's performance is largely stable for moderate truncations (4–16 modes), and interestingly at 4 modes (though k=8 captures 99% of the energy, k=4 converges upon almost 97% already) the rollout is marginally more stable than 8 modes. The extreme truncation to 2 modes and 1 mode produces a noticeable drop in accuracy, as expected, highlighting the importance of retaining a sufficient number of spectral modes to capture the essential dynamics.

Table 13: Ablation on Fourier mode truncation in TempO.

| Modes | MSE | SpectralMSE | PSNR | Pearson | MSE/time |
|---|---|---|---|---|---|
| 1 | 5.34e-02 | 3.74e-2 | 25.416 | 0.970 | 3.950e-01 |
| 2 | 3.16e-02 | 3.79e-2 | 27.695 | 0.982 | 1.668e-01 |
| 4 | 2.72e-02 | 4.00e-2 | 28.346 | 0.984 | 2.715e-02 |
| 8 | 2.91e-02 | 4.10e-2 | 28.056 | 0.983 | 3.244e-02 |
| 16 | 2.80e-02 | 8.57e-2 | 28.215 | 0.984 | 3.084e-02 |

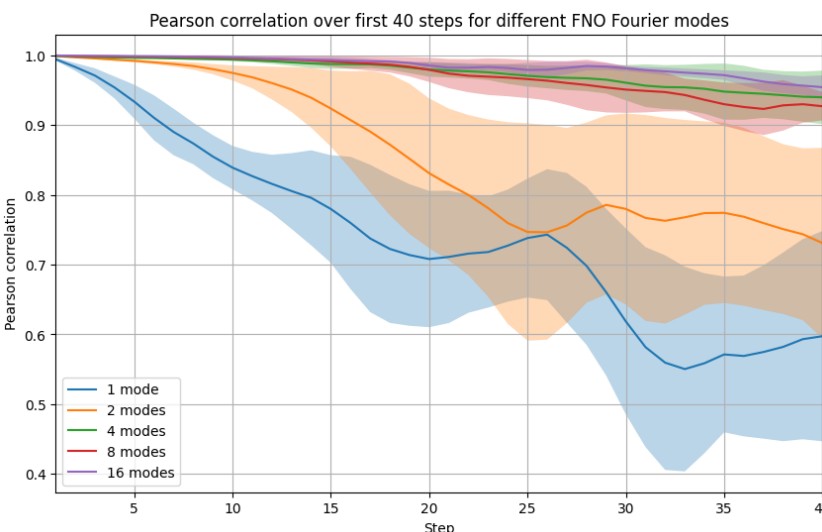

Figure 17: Ablation of varying Fourier mode truncation to evaluate sensitivity to the truncation level.

Overall, this confirms that TempO is robust to Fourier-mode selection within a reasonable range, though the best performing model is at 16 modes.

Crucially, truncation from 16 to 8 modes does not materially affect accuracy, consistent with the observation that the datasets considered have weak energy content in the highest spectral bands (Appendix G). Only extremely aggressive truncation (e.g. $m \leq 2$) produces meaningful degradation, as such settings remove both high- and mid-frequency modes required to represent the system's dynamics. Any truncation for computational requirements is also mitigated by the fact that we operate in latent space, which caps the maximum number of modes according to the resolution of the latent space.

## N    EXTENDED RESULTS FO FNO-2D ON NAVIER STOKES VORTICITY

## O    USE OF LARGE LANGUAGE MODELS (LLMS)

We acknowledge the use of ChatGPT to make suggestions on how to polish the text, correct grammar, and ensure clarity in writing. No results, code, or data were created or altered by the model.

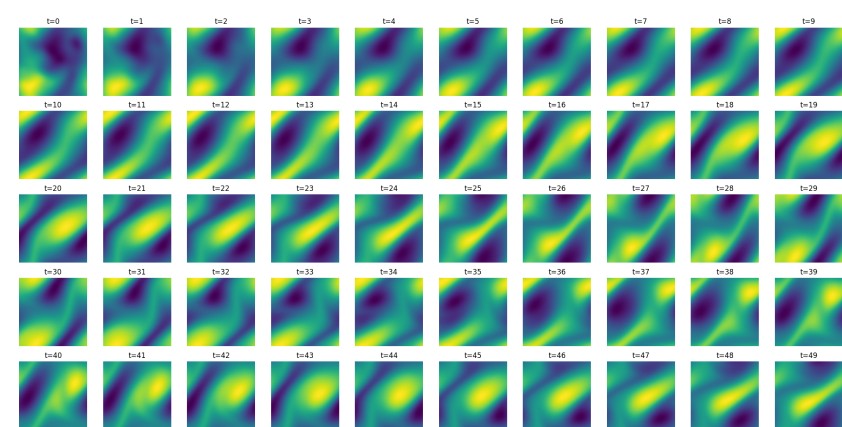

Figure 18: **Navier–Stokes Vorticity (Ground Truth).** Forecasted timeseries across 40 timesteps.

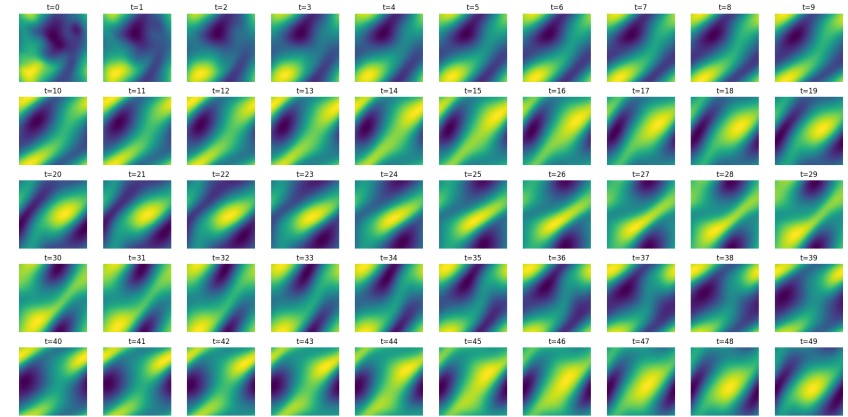

Figure 19: **Navier–Stokes Vorticity (FNO-2D).** Forecasted timeseries across 40 timesteps.

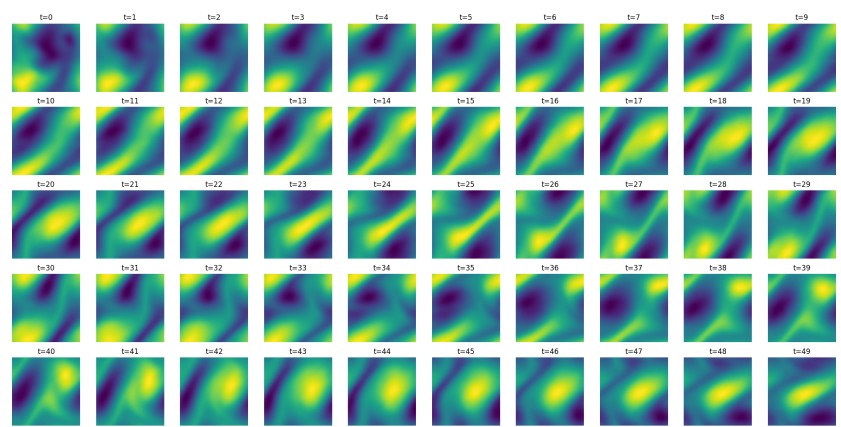

Figure 20: **Navier–Stokes Vorticity (TempO).** Forecasted timeseries across 40 timesteps.

