# OpenReview forum: "Operator Flow Matching for Timeseries Forecasting"
_ICLR.cc/2026/Conference — Submitted to ICLR 2026_

### Official Review · Reviewer_sseo · 2025-10-19

**Soundness:** 3
**Presentation:** 2
**Contribution:** 2
**Rating:** 6
**Confidence:** 2

**Summary:**

This paper introduces TempO, a novel framework for PDE forecasting based on flow matching in latent space. Unlike stochastic diffusion models, TempO leverages deterministic ODE-based sampling through a time-conditioned Fourier Neural Operator (FNO) to capture both global and local spectral dynamics. The paper provides theoretical error bounds, showing that FNOs can approximate flow fields more efficiently than Transformer- or U-Net–based samplers. Experiments are conducted on three PDE benchmarks (Navier–Stokes, Shallow Water, Reaction–Diffusion).

**Strengths:**

1. The paper is technically solid, with a clear theoretical contribution (Theorem 3.1, Proposition 3.2) establishing approximation bounds for FNO-based flow matching.
2. The idea of coupling flow matching with operator learning (FNO) is elegant and well-motivated.
3. The experimental evaluation is rigorous and diverse, covering several PDE datasets and comparing against recent ViT-based and diffusion-based baselines.

**Weaknesses:**

1. The training details (e.g., hyperparameter sensitivity, stability beyond 40-step rollouts) are under-discussed. Since the main claim is long-horizon stability, results on longer or more chaotic regimes would strengthen the argument.
2. The novelty claim could be better contextualized: related works such as Functional Flow Matching (Kerrigan et al., 2023) and Conditional Flow Matching (Tamir et al., 2024) are mentioned but not deeply contrasted with TempO in terms of scalability or architecture

**Questions:**

1. Please see the weaknesses above.
2. Could the model extend to irregular sampling or missing data, for example through Neural ODE–like continuous-time conditioning?
3. How sensitive is TempO’s performance to the number of Fourier modes or truncation level (beyond the eight-mode empirical finding)?

---

> ### Author Response · Authors · 2025-11-24
>
> We thank the reviewer for the encouraging assessment of the paper, particularly the recognition of the theoretical contribution which was then reflected by the implementation details and experimentation. We address the remaining points below.
>
> **W1: Training details and stability beyond 40 steps.**
> We appreciate this suggestion. The stability beyond 40 steps is a crucial next step to our work. We have currently tested against the PDEBench and original FNO paper (Li et al. 2021) datasets, motivated by their uptake among the community and in effort to have results for these standardised and publicly available datasets. However, we appreciate raising this point, as it highlights an important direction for future work: extending the analysis to regimes with high-frequency–dominated dynamics (e.g. turbulence at higher Reynolds numbers or multiscale physics), and to generate longer timeseries data to fully evaluate the limits of our method.
>
> That being said, in the revision we provide additional results of the MSE/time (as in the ablation in Table 5) for both Tables 2 and 3 (main quantitative results across three PDEs) as a scalar metric for the rollout error at test time, which provides an at-a-glance indicator of the stability for the experiments provided. We have additionally contextualised the importance of this stability by comparing against baseline non-flow-matching methods, and provide the revised quantitative results below for Table 2:
> | Regressor | Path   | MSE ↓     | SpectralMSE ↓ | RFNE ↓    | PSNR ↑ | Pearson ↑ | SSIM ↑ | MSE/time ↓ |
> |-----------|--------|-----------|---------------|-----------|--------|-----------|--------|------------|
> | TempO     | River  | 5.63e-02  | 3.84e-02      | 2.50e-01  | 25.19  | 0.969     | 0.786  | 2.67e-02 |
> | TempO     | Affine | 5.77e-02  | 3.98e-02      | 2.54e-01  | 25.08  | 0.968     | 0.789  | 2.91e-02 |
> | TempO     | VP     | 8.10e-02  | 5.34e-02      | 2.85e-01  | 23.61  | 0.955     | 0.731  | 2.29e-01 |
> | TempO     | VE     | 2.96e-01  | 1.73e-01      | 5.60e-01  | 17.98  | 0.821     | 0.373  | 5.02e-01 |
> | ViT       | Affine | 6.75e-02  | 4.38e-02      | 2.72e-01  | 24.40  | 0.962     | 0.758  | 8.71e-02 |
> | ViT       | River  | 6.88e-02  | 4.33e-02      | 2.73e-01  | 24.32  | 0.962     | 0.750  | 3.85e-02 |
> | ViT       | VP     | 7.77e-02  | 4.95e-02      | 2.89e-01  | 23.79  | 0.956     | 0.729  | 6.65e-02 |
> | ViT       | VE     | 1.63e+00  | 9.27e-01      | 1.35e+00  | 10.67  | 0.118     | 0.024  | 1.67e+00 |
> | U-Net     | VP     | 4.05e-01  | 3.26e-01      | 6.71e-01  | 16.62  | 0.756     | 0.323  | 2.66e-01 |
> | U-Net     | River  | 4.08e-01  | 3.28e-01      | 6.74e-01  | 16.59  | 0.752     | 0.321  | 2.79e-01 |
> | U-Net     | Affine | 4.10e-01  | 3.42e-01      | 6.76e-01  | 16.57  | 0.751     | 0.324  | 2.82e-01 |
> | U-Net     | VE     | 5.02e-01  | 3.70e-01      | 7.48e-01  | 15.68  | 0.694     | 0.263  | 2.92e-01 |
> | Baselines | FNO-2D | 6.09e-04  | 4.27e-04      | 2.54e-02  | 44.85  | 1.000     | 0.992  | 1.92e-01 |
> | Baselines | FNO-3D | 1.06e-01  | 7.34e-02      | 3.37e-01  | 22.46  | 0.945     | 0.645  | 6.37e-02 |
> | Baselines | WNO-2D | 3.72e-03  | 2.83e-03      | 6.06e-02  | 36.99  | 0.998     | 0.966  | 5.19e-01 |
> | Baselines | WNO-3D | 2.23e-01  | 1.19e-01      | 4.97e-01  | 19.21  | 0.868     | 0.452  | 2.05e-01 |
> | Baselines | U-Net  | 2.47e-03  | 1.92e-03      | 4.83e-02  | 38.772 | 0.999     | 0.976  | 1.66e+00 |

---

> > ### Author Response · Authors · 2025-11-26
> >
> > As an addendum, we provide the revised Pearson correlation per Timestep graph with mean and standard deviation, and including FNO-2D and FNO-3D results to show the error accumulation and high variance of the autoregressive method past the first 10 generated steps. We additionally provide another visualised rollout to further qualify the graph, where TempO generates a rollout capturing both high frequency detail (vortices) and high fidelity to the ground truth data as opposed to the FNO-2D which diverges despite having indisputably good next-step generation.
> >
> > https://ibb.co/album/p6BNTk

---

> ### Author Response · Authors · 2025-11-24
>
> **W2: Novelty relative to Functional and Conditional Flow Matching.**
> Thank you for noting this. We will add a more explicit comparison highlighting how TempO differs to Functional Flow Matching and Conditional Flow Matching, and how the architectural integration with FNOs enhances scalability as well as how it addresses our initial motivating problems of cumulative error and discretisation artifacts.
>
> Both Functional Flow Matching (FFM) and TempO can be formulated within the same latent interpolation and regression framework. A time-dependent velocity field $v_\theta : \mathcal{X} \times [0,1] \times \mathcal{C} \to \mathcal{X}$ is trained with the standard flow-matching objective (see Appendix B), and sampling corresponds to integrating the learned ODE $\frac{dz(t)}{dt} = v_\theta(z(t),t;\mathcal{C}),
> \qquad t\in[0,1]$.
>
> While both FFM and TempO are flow-matching ODEs in latent space, the nature and purpose of the transport differ fundamentally. In FFM, the endpoints $(Z_0,Z_1)$ are drawn independently from reference and data distributions, and integrating $v_\theta$ produces diverse samples from the target measure:
> $z(1) \sim \pi_1.$
>
> Here, the velocity field models a generative mapping over the data distribution, and the temporal component $t$ is purely an interpolation parameter without direct physical meaning. Conditional FFM methods (e.g., Tamir et al., 2024) introduce context $\mathcal{C}$ to guide sampling but still produce stochastic outputs rather than deterministic forecasts.
>
> In contrast, TempO constructs $(Z_0,Z_1)$ from latent encodings of a short temporal context and its future, with the velocity field conditioned on $\mathcal{C} = \{u_0,\tau\}$. Integrating the learned flow produces a single, deterministic trajectory corresponding to the evolution of the specific initial condition $u_0$ over forecast horizon $\tau$:
> $z(1) = \text{ODE}\big(z(0)=Z_0; v_\theta(\cdot, t;\mathcal{C})\big),
> \qquad \Phi_\theta(u_0,\tau) = \mathcal{D}(z(1)).$
> Thus, TempO functions as a conditional, operator-valued map: each integration produces a trajectory consistent with the underlying PDE dynamics rather than sampling a distribution over possible outcomes.
>
> A key architectural distinction is TempO’s channel folding. This allows the vector field regressor (the FNO backbone) to learn stationary intrinsic-time dynamics $t$ that define local spatial evolution, while the extrinsic forecast horizon $\tau$ is explicitly provided via conditioning $\mathcal{C} = \{u_0, \tau\}$ and modulates the spatial operator through a learned temporal flow. This disentangles $t$ (within the flow matching integration step) from $\tau$ (being the PDE timestep), which is explicitly provided as part of the conditioning tuple $\mathcal{C} = \{z_\tau, \Delta\}$. This channel-folded design enables efficient, high-dimensional latent forecasting while preserving the interpretability and determinism of classical PDE evolution operators.
>
> Consequently, TempO is built  by design to scale to long time horizons and higher-dimensional latent spaces with improved stability and spectral fidelity compared to both FFM and conditional FM, while maintaining rigorous adherence to the underlying physical dynamics.

---

> ### Author Response · Authors · 2025-11-24
>
> **Q1: Irregular sampling or missing data.**
> This is an excellent question! TempO can absolutely handle such settings in theory without much adjustment. There are two regimes of missing data which I consider: missing data/irregular sampling in the spatial domain, and irregular spacing in time.
>
> *Spatial irregularity:* For irregularly sampled spatial points, a masking strategy can be employed to indicate which points are observed. The latent encoding $f_\phi(x)$ can then be computed only over available points, and the FNO-based vector field regressor $v_\theta$ can be conditioned to ignore or interpolate unobserved locations, analogous to standard graph- or point-based neural operators. The Fourier backbone would in theory aid in this sparse sensing regime, which is further substantiated by a very recent paper (to be added to related works in the revision) "Stochastic Process Learning via Operator Flow Matching" (Shi et al. 2025) which extends Functional Flow Matching to accommodate the probability distribution of any set of points and allows for mathematically tractable functional regression at new locations, providing both mean predictions and density estimates. The extension of this to our deterministic trajectories over time is part of our current exploration as a natural next step of our investigation, and we apprerciate the interest in how to extend our method to real world scenarios. Will be added as future work in the updated manuscript.
>
> *Temporal irregularity:* For non-uniform timesteps, the conditioning and reference embeddings can incorporate the actual timestamp information. The temporal offset $\Delta$ is computed based on the real timestamps rather than fixed indices, allowing the latent velocity field $v_\theta(z_t, t; \mathcal{C})$ to evolve over arbitrary continuous time intervals. In this sense, TempO behaves like a Neural ODE in latent space, integrating the learned flow over continuous time. This ensures that the model can generate forecasts even with irregularly spaced observations, while preserving the deterministic, operator-valued nature of the trajectory.
>
> Together, these adaptations enable TempO to generalize to a wide range of real-world scenarios with missing or unevenly sampled data, leveraging its continuous-time latent flow framework.
>
> **Q2: Sensitivity to Fourier modes.**
>
> We thank the reviewer for this question. To evaluate the sensitivity of TempO to Fourier mode truncation, we conducted an ablation study across different numbers of modes: 16, 8, 4, 2, and 1 (the most extreme truncation). These results are included in the Appendix.
>
> The study shows that TempO’s performance is largely stable for moderate truncations (4–16 modes), and interestingly at 4 modes (though k=8 captures 99\% of the energy, k=4 converges upon almost 97\% already) the rollout is marginally more stable than 8 modes. The extreme truncation to 2 modes and 1 mode produces a noticeable drop in accuracy, as expected, highlighting the importance of retaining a sufficient number of spectral modes to capture the essential dynamics. Overall, this confirms that TempO is robust to Fourier-mode selection within a reasonable range, though the best performing model is at 16 modes. Any truncation for computational requirements is also mitigated by the fact that we operate in latent space, which caps the maximum number of modes according to the resolution of the latent space.
>
> | Modes | MSE      | SpectralMSE | PSNR   | Pearson | MSE/time  |
> | ----: | -------- | ----------- | ------ | ------- | --------- |
> |     1 | 5.34e-02 | 3.74e-02    | 25.416 | 0.970   | 3.950e-01 |
> |     2 | 3.16e-02 | 3.79e-02    | 27.695 | 0.982   | 1.668e-01 |
> |     4 | 2.72e-02 | 4.00e-02    | 28.346 | 0.984   | 2.715e-02 |
> |     8 | 2.91e-02 | 4.10e-02    | 28.056 | 0.983   | 3.244e-02 |
> |    16 | 2.80e-02 | 8.57e-02    | 28.215 | 0.984   | 3.084e-02 |
>
> We thank the reviewer again for the supportive feedback and helpful questions! The feedback provided has helped us consider more critically how to present the novelty of our method, and we hope the revision has captured our contributions more clearly.

---

### Official Review · Reviewer_txKP · 2025-10-23

**Soundness:** 1
**Presentation:** 2
**Contribution:** 2
**Rating:** 2
**Confidence:** 3

**Summary:**

This paper tackles long-horizon forecasting for high-dimensional PDE dynamics , proposing Tempo, a novel latent flow matching model. Tempo's core innovation is using a time-conditioned Fourier Neural Operator (FNO) as its velocity field regressor. The model operates in a latent space, employing sparse conditioning and channel folding for efficient 3D spatiotemporal processing. The authors provide theoretical justification that this FNO-based design is asymptotically more parameter-efficient than sampler-based architectures like Transformers or U-Nets. Experimentally, Tempo outperforms SOTA baselines on three PDE datasets, demonstrating highly stable long-horizon forecasting where competitors fail , while also being significantly more parameter-efficient

**Strengths:**

* Proposes **Tempo**, a novel model combining latent flow matching with an FNO regressor, which is well-motivated by aligning the FNO's spectral bias with continuous PDE dynamics.

* Provides a theoretical analysis (Theorem 3.1, Prop 3.2) to justify the architecture, suggesting FNOs can achieve a target accuracy with asymptotically fewer parameters than sampler-based models like U-Nets or Transformers.

* Demonstrates highly stable 40-step autoregressive forecasting on the NS-w dataset. This significantly outperforms ViT and U-Net baselines, which show clear degradation.

* The model is highly parameter-efficient. It also shows superior sampling efficiency, requiring the fewest Number of Function Evaluations (NFEs) for inference.

**Weaknesses:**

* The paper suffers from significant clarity issues. It lacks a high-level overview of the proposed method, making it difficult to grasp the core components. The writing style relies on overly long and convoluted sentences, hindering readability. The overall structure feels disjointed, making the paper's narrative hard to follow.

* The paper's core motivation—that existing methods fail due to "cumulative errors and discretisation artifacts"—is not sufficiently substantiated. This claim is presented as a given, but the paper lacks the necessary citations or analysis to support it.

* The connection between the stated problem and the proposed solution is weak. The paper does not adequately explain *why* flow matching is the right choice to mitigate "cumulative errors" or *how* the FNO architecture specifically addresses "discretisation artifacts" better than other regressors. The design choices feel disconnected from the initial problem statement.

* The methodological novelty is unclear. The primary components, Flow Matching and FNO, are both well-established. The paper fails to clearly articulate what makes their specific combination (Tempo) a significant and novel contribution beyond a straightforward engineering application.

* The empirical results are not fully convincing.
    * The performance gains over the ViT and U-Net baselines are marginal in several next-step prediction tasks. The strong long-rollout performance is only demonstrated on one dataset (NS-w).
    * The baseline comparison is insufficient. The paper omits direct comparisons against the original FNO (which should be a key baseline/ablation) and other strong operator learning models (e.g., WNO, DeepONet), making it difficult to properly contextualize the results.

**Questions:**

1.  The paper's motivation hinges on addressing "cumulative errors" and "discretisation artifacts." Could the authors elaborate on the explicit mechanism by which the flow matching framework and the FNO's spectral bias inherently mitigate these specific error types, in a way that standard autoregressive U-Nets or Transformers do not?

2.  The experimental comparison focuses on ViT and U-Net regressors *within* the flow matching setup. Could the authors provide a comparison against a standard autoregressive FNO (or other strong neural operators like WNO/DeepONet)? This seems crucial to disentangle the benefits of the proposed flow matching framework from the known benefits of the FNO architecture itself.

See the above weaknesses for further details.

---

> ### Author Response · Authors · 2025-11-24
>
> We thank the reviewer for the detailed and technically insightful feedback. We appreciate the recognition of our broad baseline set, the theoretical motivation for using FNOs as denoisers, and some of our highlighted results. More importantly, we appreciate your critical feedback which resulted in productive revisions! Below we address the concerns raised.
>
> **W1: Clarity of writing.**
>
> Thank you for your feedback. We will improve the high-level overview, clarify the method, and make longer sentences more concise!
>
> The key methodological contributions are as follows. We will revise the manuscript to highlight these points more clearly, including using better formatting to avoid longer blocks of text.
>
> * *Continuous-time latent evolution:* TempO learns a time-conditioned latent velocity field
> $v_\theta(z,t)$ defining $\partial_t z = v_\theta(z,t)$, with $z = f_\phi(x)$.
> This avoids error accumulation from stepwise autoregressive forecasting.
>
> * *Operator-valued map / PDE parallel:* Integrating the latent flow yields a deterministic trajectory $z(1) = \text{ODE}(z(0)=Z_0; v_\theta(\cdot, t;\mathcal{C})), \quad \Phi_\theta(u_0,\tau) = \mathcal{D}(z(1))$,
>   mimicking classical PDE evolution operators that deterministically transport initial states forward in time.
>
> * *Channel folding for temporal disentanglement:* Folding batch and channel dimensions,
>   $ u \in \mathbb{R}^{B \times C \times T \times H \times W} \mapsto u' \in \mathbb{R}^{(B \cdot C) \times T \times H \times W}$,
>
>   lets the FNO learn intrinsic-time spatial dynamics $t$, while the extrinsic forecast horizon $\tau$ is provided via conditioning $\mathcal{C} = {u_0, \tau}$, preventing premature encoding of full physical time.
>
> * *Spectral inductive bias and stability:* The FNO latent regressor $v_\theta(z,t)$ captures long-range correlations and suppresses high-frequency aliasing:
>   $$
>   \mathcal{K}(z) = \mathcal{F}^{-1}(R_\theta(k)\mathcal{F}(z)).
>   $$
>   Unlike FFM, which applies FNO directly in data space and treats $t$ as a flow parameter, TempO leverages latent space structure for stable long-horizon forecasts.
>
> * *Sparse temporal conditioning per Davtyan et al. 2023.* Conditioning on a reference $z_T$ and offset embedding $z_\tau$ with $\Delta=T-\tau$ enables parameter-efficient modeling while retaining long-range temporal dependencies.
>
> These elements together allow TempO to efficiently learn deterministic, operator-valued latent flows that capture complex PDE evolution over long horizons.
>
> **W5: Empirical results (rollout error, baseline comparisons).**
>
> We agree that rollout error is central for PDE surrogate modeling. In response, we are including averaged MSE/time (over the rollout of 40 steps) to the results table to establish a scalar comparison which, while not as comprehensive as the full trajectory, can indicate the performance of each model not only for next step prediction as we show currently, but also across the generated rollout.
>
> We also appreciate that including autoregressive baselines would help calibrate performance. We have generated comparisons to vanilla FNO, WNO, and UNet models to address this gap, and will provide this in the revision; the updated results table is shown under **Q2**.
>
> In addition, we have revised the rollout in Figure 1 to include the mean and std averaged over 10 randomly selected trajectories for each method, and include a baseline FNO rollout for a more rigorous comparison between methods.

---

> ### Author Response · Authors · 2025-11-24
>
> **W2: Cumulative errors and discretisation artifacts**
>
> This is a sharp observation and we thank you for your feedback. We thank the reviewer for this insightful point and have strengthened both our empirical and conceptual justification. First, we now include additional results for three standard autoregressive baselines (FNO-2D, WNO-2D, and U-Net), all of which clearly exhibit the well-documented phenomenon of error amplification under rollout. This aligns with a large body of prior work: Bengio et al. (2015) describe exposure bias and the accumulation of compounding errors in autoregressive predictors; Sanchez-Gonzalez et al. (2020) and Brandstetter et al. (2022) explicitly report that even strong physics simulators deteriorate rapidly under long-horizon rollout; and Li et al. (2021) show that discretisation-dependent architectures such as CNNs and classical regression models introduce mesh sensitivity and artifacts that accumulate across iterative predictions. Coarse-graining and tokenisation effects are also known to introduce aliasing and truncation errors that propagate temporally (Stachenfeld et al., 2022). More formally, Kovachki et al. (2023) demonstrate that neural networks approximate discretised operators, not the underlying continuous operators themselves, implying that discretisation-induced approximation errors enter at training time and amplify during autoregressive evolution.
>
> We find that cumulative errors and discretisation artifacts are pervasive and well-studied limitations of autoregressive PDE forecasting methods, and our motivation is to explore continuous-time generative approaches that reduce discretisation sensitivity, and do not propagate errors across steps. We will update the introduction and some discussion to more explicitly ground this motivation in prior work, and appreciate this important point of clarification.
>
> **W3, Q1: Relationship between motivation and method**
>
> This is indeed a critical question and we will revise our methodology section to reflect a more rigorous relationship between our motivation and the method, as also pointed out by W1.
>
> In TempO, cumulative autoregressive error is mitigated because the model does not apply a discrete recurrence of the form $(u_{t+\Delta t} = f_\theta(u_t))$. Instead, it learns a continuous-time evolution operator
>
> $\hat{u}*t = \Phi*\theta(u_0,t), \qquad \partial_t u = v_\theta(u,t)$,
>
> so prediction errors do not propagate forward in time; the conditioning is always anchored on the initial state $(u_0)$. This differs fundamentally from U-Nets/Transformers whose local, stepwise maps amplify errors through repeated application.
>
> Regarding discretisation artifacts, the flow-matching formulation avoids finite-difference–like updates $(u_{t+\Delta t}=u_t+\Delta t,g_\theta(u_t))$, while the FNO backbone
>
> $\mathcal{K}(u)=\mathcal{F}^{-1}(R_\theta(k),\mathcal{F}(u))$
>
> provides a global spectral operator that suppresses high-frequency aliasing and grid-scale noise. The combination of (i) continuous-time modeling and (ii) FNO’s spectral filtering yields improved stability relative to standard autoregressive architectures.
>
> **W4: Clarification on Novelty**
>
> We thank the reviewer for this comment. While both Flow Matching and FNO are well-established individually, TempO combines them in a manner specifically designed for long-horizon PDE forecasting. In particular, TempO disentangles intrinsic latent dynamics from extrinsic forecast time using channel folding and conditioning on $(u_0, \tau)$, enabling the model to learn a deterministic, operator-valued evolution map that produces stable trajectories. Unlike standard flow matching, which transports distributions between reference and data measures, TempO defines $(x_0, x_1)$ via latent encodings of short temporal contexts, thereby generating conditional trajectories that mitigate cumulative errors in autoregressive rollouts. The FNO backbone further introduces spectral bias and global correlations, allowing the model to maintain long-horizon accuracy with fewer parameters than comparable architectures. In short, the novelty of TempO lies in the integration of flow matching with a spectral operator backbone, intrinsic/extrinsic time disentanglement via channel folding, and conditioning on initial states and forecast horizons, producing deterministic, operator-informed trajectories, an approach not realized in prior work. TempO draws inspiration from classical evolution operators in PDE theory, treating the initial state as a deterministic latent trajectory that is continuously transported forward, thereby embedding the underlying physical structure directly into the generative model.

---

> ### Author Response · Authors · 2025-11-24
>
> **Q2: Experimental comparisons against standard baseline models**
>
> Thank you for this important observation. We agree that comparisons to deterministic PDE surrogates such as FNO, WNO, DeepONet, and diffusion-based PDE models are needed to help contextualize performance beyond flow-matching baselines. To this end, we have carried out further extensive experiments. In the revised version, we extend the results to compare against pure U-Net, FNO and WNO autoregressive models as well as 3D convolutional extensions the latter two neural operators as suggested by their respective authors, which additionally convolve over time. Hyperparameter settings chosen per the original papers (Li et al. 2021, Tripura et al. 2022) for the Navier Stokes vorticity dataset, and we provide hyperparameter and additional training details in a new appendix section.
>
> These additional results confirm our hypothesis that while purpose built models such as the FNO-2D and WNO-2D (and, indeed, the U-Net) outperform TempO and other flow matching methods for next step prediction, they fail at the rollout stage with significantly higher MSE/time. We also add the baseline model with the best rollout, FNO-3D to Figure 1 to compare directly with TempO and the top flow matching competitors. A few observations: the FNO3D is the only model where the first timestep is not necessarily the most accurate, which is indicative of its treatment convolving over all 3 spatiotemporal dimensions at once; the FNO2D baseline exhibits the expected degradation as error accumulates (also indicated by the large variance seen across multiple generations) despite having the best next-step performance. We also note that Figure 1 has been modified to show the mean and standard deviation averaged across 10 randomly selected initial conditions from the withheld test set.
>
> | Regressor | Path   | MSE ↓     | SpectralMSE ↓ | RFNE ↓    | PSNR ↑ | Pearson ↑ | SSIM ↑ | MSE/time ↓ |
> |-----------|--------|-----------|---------------|-----------|--------|-----------|--------|------------|
> | TempO     | River  | 5.63e-02  | 3.84e-02      | 2.50e-01  | 25.19  | 0.969     | 0.786  | 2.67e-02 |
> | TempO     | Affine | 5.77e-02  | 3.98e-02      | 2.54e-01  | 25.08  | 0.968     | 0.789  | 2.91e-02 |
> | TempO     | VP     | 8.10e-02  | 5.34e-02      | 2.85e-01  | 23.61  | 0.955     | 0.731  | 2.29e-01 |
> | TempO     | VE     | 2.96e-01  | 1.73e-01      | 5.60e-01  | 17.98  | 0.821     | 0.373  | 5.02e-01 |
> | ViT       | Affine | 6.75e-02  | 4.38e-02      | 2.72e-01  | 24.40  | 0.962     | 0.758  | 8.71e-02 |
> | ViT       | River  | 6.88e-02  | 4.33e-02      | 2.73e-01  | 24.32  | 0.962     | 0.750  | 3.85e-02 |
> | ViT       | VP     | 7.77e-02  | 4.95e-02      | 2.89e-01  | 23.79  | 0.956     | 0.729  | 6.65e-02 |
> | ViT       | VE     | 1.63e+00  | 9.27e-01      | 1.35e+00  | 10.67  | 0.118     | 0.024  | 1.67e+00 |
> | U-Net     | VP     | 4.05e-01  | 3.26e-01      | 6.71e-01  | 16.62  | 0.756     | 0.323  | 2.66e-01 |
> | U-Net     | River  | 4.08e-01  | 3.28e-01      | 6.74e-01  | 16.59  | 0.752     | 0.321  | 2.79e-01 |
> | U-Net     | Affine | 4.10e-01  | 3.42e-01      | 6.76e-01  | 16.57  | 0.751     | 0.324  | 2.82e-01 |
> | U-Net     | VE     | 5.02e-01  | 3.70e-01      | 7.48e-01  | 15.68  | 0.694     | 0.263  | 2.92e-01 |
> | Baselines | FNO-2D | 6.09e-04  | 4.27e-04      | 2.54e-02  | 44.85  | 1.000     | 0.992  | 1.92e-01 |
> | Baselines | FNO-3D | 1.06e-01  | 7.34e-02      | 3.37e-01  | 22.46  | 0.945     | 0.645  | 6.37e-02 |
> | Baselines | WNO-2D | 3.72e-03  | 2.83e-03      | 6.06e-02  | 36.99  | 0.998     | 0.966  | 5.19e-01 |
> | Baselines | WNO-3D | 2.23e-01  | 1.19e-01      | 4.97e-01  | 19.21  | 0.868     | 0.452  | 2.05e-01 |
> | Baselines | U-Net  | 2.47e-03  | 1.92e-03      | 4.83e-02  | 38.772 | 0.999     | 0.976  | 1.66e+00 |
>
> We thank the reviewer again for the constructive feedback and believe these clarifications and additions will significantly strengthen the paper. We found the criticism insightful and resulted in more careful, concise, and targeted discussion of our methodology and its placement within the literature.

---

> > ### Comment · Reviewer_txKP · 2025-11-25
> >
> > Thanks for the response. After carefully reviewing the new results, I still have significant reservations regarding the core claims of the paper.
> >
> > First, regarding the mitigation of cumulative errors, I remain skeptical. While "continuous-time latent evolution" avoids explicit autoregressive steps, the flow matching integration process itself inevitably introduces numerical errors via the ODE solver. It is not theoretically or empirically obvious that the integration error over the latent trajectory is strictly bounded below the accumulation error of a well-tuned autoregressive model. Consequently, the reliance on "continuous-time modeling" as a fundamental solution feels somewhat trivial without stronger justification.
> >
> > Second, the novelty argument remains unconvincing. As the authors acknowledged, both Flow Matching and FNO are well-established components. Combining them for PDE forecasting appears to be a relatively straightforward engineering adaptation rather than a distinct methodological breakthrough. I struggle to see a unique insight in the proposed combination that goes beyond standard architectural blending.
> >
> > Most importantly, the new experimental data exposes a worrying trade-off that undermines the motivation of "high-fidelity" modeling. The results show that Tempo is nearly two orders of magnitude worse than FNO-2D in capturing immediate dynamics (Next-step MSE: ~3.84e-2 vs ~4.27e-4). This raises a critical question: Does Tempo achieve "stability" simply by learning a blurred or averaged representation that refuses to diverge, effectively sacrificing high-frequency physical fidelity for visual stability? The significantly higher RFNE of Tempo compared to FNO-2D supports this suspicion. If the model maintains stability merely by smoothing out complex dynamics, its utility for precise scientific forecasting is questionable.

---

> > > ### Author Response · Authors · 2025-11-26
> > >
> > > Thank you for your further discussion!
> > >
> > > **Baseline FNO Results Clarification**: We appreciate that just the metrics for baseline models do not fully capture the performance. Your concern that TempO blurs for a stable solution can clearly be addressed by both Figure 1 in accompanying text where we highlight its ability to capture detailed vortices which are high frequency phenomena, and additional results in the appendix section of the full rollout over time. In fact, we see the sacrifice of some high-frequency physical fidelity in the FNO case and provide both the revised Pearson per timestep graph and the full visualised rollout in the below anonymised link.
> > >
> > > https://ibb.co/album/p6BNTk
> > >
> > > We appreciate that the next-step RFNE, like the next-step MSE, resulting from the FNO model performs better than TempO; however, the RFNE averaged over the 40 generated timesteps (RFNE/time) are 0.48 (FNO-2D), 0.27 (FNO-3D), and 0.16 (TempO) respectively, showing that although the next step performance of the FNO is indisputably strong, it does not mitigate the error accumulated in the autoregressive rollout.
> > >
> > > **Autoregressive vs flow matching cumulative error**: We thank the reviewer for this important point. Below we give a short, semi-rigorous comparison showing how errors propagate in (A) autoregressive discrete models and (B) continuous-time latent integration (TempO). The argument clarifies assumptions under which integration of a smooth learned vector field can avoid the multiplicative error amplification characteristic of autoregressive rollouts.
> > >
> > > The reviewer is correct that numerical integration introduces error. However, the theoretical results of Benton et al. (2024, Error Bounds for Flow Matching Methods) show that the error accumulation for flow matching is qualitatively different from the multiplicative accumulation characteristic of autoregressive rollouts.
> > >
> > > In particular, if the learned drift satisfies the standard flow matching assumptions
> > >
> > > $
> > > \int_{0}^{1}\mathbb{E}\big|v_\theta(X_t,t)-v_X(X_t,t)\big|^2,dt
> > > ;\le; \varepsilon^2,
> > > \qquad
> > > |v_\theta(\cdot,t)|_{\mathrm{Lip}} \le L_t,
> > > $
> > >
> > > then the 2-Wasserstein error between the learned flow and the true flow is bounded polynomially in $\varepsilon$ and depends on the spatial Lipschitz constant $L_t$. Under these assumptions, the global error grows as
> > >
> > > $
> > > W_2(\pi_1^\theta,\pi_1)
> > > ;\le;
> > > \exp!\Big(\int_0^1 L_t,dt\Big),\varepsilon,
> > > $
> > >
> > > which corresponds to the classical Grönwall-type stability estimate for ODEs.
> > >
> > > By contrast, an autoregressive predictor with one-step Jacobian $J_t$ accumulates error according
> > >
> > > $
> > > |e_T|
> > > ;\le;
> > > \Bigg(\prod_{t=1}^{T}|J_t|\Bigg),|e_0|
> > > ;+;
> > > \sum_{k=1}^{T}
> > > \Bigg(\prod_{t=k+1}^{T}|J_t|\Bigg),\delta,
> > > $
> > >
> > > where $\delta$ is the one-step prediction error. This yields *multiplicative* amplification whenever $|J_t| > 1$. Even if $\delta$ is matched across both models, the exponential factor $\prod_t |J_t|$ dominates long-range forecasts.
> > >
> > > Thus, the distinction is not that ODE integration is error-free, but that flow matching inherits the stability guarantees of smooth ODEs with Lipschitz drifts, whereas autoregressive rollouts inherit the instability of repeated Jacobian multiplication. The theoretical results therefore show that flow matching errors tend to remain controlled under the assumptions naturally encouraged by the FM objective, while autoregressive errors may grow exponentially even with comparable one-step errors. Of course, we acknowledge that how well an autoregressive model and how well a flow matching model are trained will impact the results, and further analysis including of the sensitivity of TempO can be added to future work as well.
> > >
> > > **Novelty**: We emphasize that TempO is inspired by classical PDE evolution operators and is designed specifically for long-horizon PDE forecasting. Its novelty lies in a specific operator-level reinterpretation of conditional flow matching for PDE evolution, enabled by practical elements such as the usage of the FNO as a vector field regressor. TempO uses it as the spatial component of an operator-valued vector field, additionally motivated by our theoretical work proving that an FNO-inspired regressor can achieve an upper bound on approximation error for flow matching models compared to the accuracy achievable by sampler-based method. This pairing yields a continuous-time model with global receptive fields, enabling stable long-horizon rollouts that autoregressive and time-convolutional FNOs cannot reliably achieve. Our ablations show that neither FM alone nor FNO alone achieves this behavior. The framing of using flow matching to learn the latent temporal evolution operator that deterministically maps an initial PDE state to a later one is also a conceptual shift that yielded promising results for stable temporal rollouts as shown in our experiments.

---

### Official Review · Reviewer_CwD2 · 2025-10-28

**Soundness:** 2
**Presentation:** 3
**Contribution:** 2
**Rating:** 4
**Confidence:** 3

**Summary:**

The authors propose using FNO as the denoiser during flow matching, motivated by some theoretical insight. When compared to ViT or Unet based denoisers across different noise schedules, the method works well and seems to improve on 2D benchmarks.

**Strengths:**

- The authors present a variety of baselines, including different noise schedules and backbones.
- The theoretical motivation for using FNO as a denoiser is apparent.
- The performance gains are good, with consistent gains across different noise schedules/datasets/baselines.

**Weaknesses:**

- I’m having a hard time differentiating this work from prior work (Functional Flow Matching https://arxiv.org/pdf/2305.17209), where an FNO is also used as the denoiser and an OT schedule is used. Is the novelty using sparse conditioning and channel folding? The additional experiments to a longer horizon are also somewhat lacking (an ablation in Table 5 for some unspecified dataset).
- Reporting next-step error is a good start, but in general, most works are concerned with rollout error since the main driver of error in neural PDE surrogates is autoregressive error accumulation. MSE/time is shown in Table 5, but it would be good to see this for all models (backbones + noise schedules), since it is more informative than single-step error.
- There seems to be better spectral performance at the highest frequency bands for ViT-based models, which may be related to FNO’s mode truncation. This might be more relevant for spectral accuracy in more complex systems (turbulence, multiscale phenomena)
- The ablation in Table 5 on the effect of context length on accuracy could be expanded on. Is the model with a sequence length of 25 predicting 15 unseen frames and the model with sequence length of 2 predicting 38 unseen frames?
    - This also seems to contradict prior results (https://arxiv.org/pdf/2507.02608,  https://arxiv.org/abs/2111.13802) that suggest that the context length either does not have an effect on rollout error or harms it.
- Some comparisons to deterministic models (FNO/Unet) for SWE/NS/RD would be beneficial just to calibrate what is the baseline performance for neural surrogates on these common systems.

**Questions:**

- There could be a few more relevant works (related to flow matching for PDEs) that could be cited:
    - Latent flow matching (https://arxiv.org/abs/2503.22600)
    - Flow matching for PDEs (https://arxiv.org/abs/2506.08604)
    - Using FNO as a Denoiser (https://arxiv.org/abs/2302.07400)
- What dataset is used in Table 5?
- Is Figure 1 generated with a single trajectory or averaged across the validation set?
- Not a problem, but there seem to be a lot of red references to lead nowhere.
- The performance of the UNet denoiser seems to be very poor based on qualitative observations (Figure 6), but since the NS dataset has been standard for quite some time now, there is a lot of prior work that shows that vanilla FNO/Unet can approximate the system well + even more challenging systems (https://arxiv.org/abs/2209.15616, https://arxiv.org/abs/2309.01745).
    - This isn’t explicitly an issue, but just curious that Unet struggles so much.
- Also not a clear issue, but the use of a transformer as a diffusion/flow matching backbone is very ingrained in modern machine learning, not only in image generation, but also in PDEs. There is a lot of prior work that uses this paradigm successfully, so suggesting an alternate and expecting people to adopt it would need to include a very rigorous set of experiments, likely on more challenging PDEs/benchmarks (https://arxiv.org/abs/2412.00568).
    - Perhaps a thought experiment would be: "If I am building a large, latent generative model, would I use a transformer or FNO as the backbone?" How would you convince someone of one method or another?

---

> ### Author Response · Authors · 2025-11-24
>
> We thank the reviewer for the detailed and technically insightful feedback! We particularly appreciate the recognition of our broad baseline set, the motivation for using FNOs as denoisers, and the consistency of performance across noise schedules and datasets. Below we address the concerns raised.
>
> **W1: Differentiation from Functional Flow Matching.**
> We appreciate the question and that there are similarities in the components used for both methods. While FFM defines flows between functions, TempO seeks to learn the flows between temporal evolutions of functions, disentangling generative time (the intrinsic integration from base distribution to target distribution) from physical time (the extrinsic time between each frame of a temporal PDE).
>
> Both Functional Flow Matching (FFM) and TempO can be written under the same latent interpolation and regression framework. A time-dependent velocity field
>
> $v_\theta : \mathcal{X} \times [0,1] \times \mathcal{C} \to \mathcal{X}$
>
> is trained by the standard flow-matching objective (Appendix B), and sampling corresponds to integrating the learned ODE
>
> $
> \frac{dz(t)}{dt} = v_\theta(z(t),t;\mathcal{C}),
> \qquad t\in[0,1].
> $
>
> While both FFM and TempO are formulated as flow-matching ODEs in latent space, crucially the nature of the transport differs.
> FFM draws endpoints $(Z_0,Z_1)$ independently from reference and data distributions and integrates $v_\theta(z,t)$ to produce \emph{diverse samples} from the data measure:
> $
> z(1) \sim \pi_1.
> $
> In contrast, TempO constructs $(Z_0,Z_1)$ from latent encodings of a short temporal context and its future, and conditions the velocity field on $\mathcal{C} = \{u_0,\tau\}$.
> Integrating the learned flow thus produces a \emph{single, deterministic trajectory} in latent space corresponding to the evolution of the specific initial condition $u_0$ over horizon $\tau$:
> $
> z(1) = \text{ODE}\big(z(0)=Z_0; v_\theta(\cdot, t;\mathcal{C})\big),
> \qquad \Phi_\theta(u_0,\tau) = \mathcal{D}(z(1)).
> $
> In this sense, TempO still performs a transport from the initial latent “point” to its evolved state, but it does not generate a distribution of outcomes; each integration produces a trajectory consistent with the underlying PDE dynamics.
> This makes TempO effectively a conditional, operator-valued map, in contrast to the unconditional generative transport learned by FFM.
>
> Additionaly, the channel folding in TempO plays a crucial role in disentangling intrinsic spatial dynamics from extrinsic time evolution. Specifically, the vector field regressor learns stationary intrinsic-time dynamics $t$ that define local spatial evolution, while the extrinsic forecast horizon $\tau$ is explicitly provided via conditioning $\mathcal{C} = \{u_0, \tau\}$ and modulates the spatial operator through a learned temporal flow.
> This allows the vector field regressor (the FNO backbone) to learn stationary intrinsic-time dynamics $t$ that define local spatial evolution, while the extrinsic forecast horizon $\tau$ is explicitly provided via conditioning $\mathcal{C} = \{u_0, \tau\}$ and modulates the spatial operator through a learned temporal flow. This disentangles $t$ (within the flow matching integration step) from $\tau$ (being the PDE timestep), which is explicitly provided as part of the conditioning tuple $\mathcal{C} = \{z_\tau, \Delta\}$. This channel-folded design enables efficient, high-dimensional latent forecasting while preserving the interpretability and determinism of classical PDE evolution operators.
> In contrast to a 3D FNO that convolves physical time as an additional input dimension, this channel-folded representation prevents the intrinsic vector field from encoding the full physical-time evolution, avoiding unwanted periodicities or entanglement between space and time. As a result, the model learns an operator-valued mapping where each extrinsic time $\tau$ selects an appropriate latent spatial operator
> to deterministically evolve the PDE state.
>
> **W2: Rollout error beyond next-step prediction.**
> We agree that rollout error is central for PDE surrogate modeling. In response, we will include the MSE/time per your suggestion as additional rollout results in the revision for both Tables 2 and 3. A revised Table 2 is presented below in **W5**, as results have been produced for both comparisons to standard baseline models as well as the MSE/time.

---

> ### Author Response · Authors · 2025-11-24
>
> **W3: High-frequency spectral behavior.**
> We thank the reviewer for this observation. The spectral behaviour of ViT-based regressors versus FNO-based regressors at high frequencies would be related to mode truncation and aliasing; however, at the resolutions datasets shown below, we  have not needed to truncate the FNO modes and therefore the observed differences at the extreme high-frequency end are not caused by explicit spectral cut-offs.
>
> Nevertheless, we agree that understanding the role of high-frequency modes is important, especially for more turbulent or strongly multiscale regimes. To contextualise the reviewer’s comment, Appendix G presented a classical spectral-truncation analysis of the ground-truth Navier–Stokes vorticity fields. This analysis shows that:
> (1) the cumulative energy of the system is overwhelmingly concentrated in low and mid frequencies;
> (2) reconstruction MSE drops rapidly once only a modest number of modes are retained; and
> (3) the highest-frequency bands contribute a very small fraction of the total energy, even under unstructured truncation.
>
> Thus, for the regimes considered in this paper, the spectral content in the top frequency bands is extremely weak, which explains why differences between architectures in these bands have negligible influence on physical-space metrics.
>
> We also evaluate TempO on the 2D Reaction–Diffusion system, which involves two nonlinearly coupled fields with sharper gradients and emergent spatiotemporal patterns. In this RD dataset, higher-frequency spectral components carry more energy due to localized wavefronts and spiral structures. Our results indicate that TempO maintains good fidelity even in these regimes, partially addressing the reviewer’s concern about architectural spectral biases and highlighting TempO’s ability to capture more complex, multiscale dynamics.
>
> We appreciate the reviewer raising this point, as it highlights an important direction for future work: extending the analysis to Navier Stokes datasets with high-frequency–dominated dynamics (e.g. turbulence at higher Reynolds numbers or multiscale physics), where architectural inductive biases in spectral propagation will play a stronger role.
>
> **W4: Context-length ablation clarity.**
> We acknowledge that the context-length configurations in Table 5 need clearer specification. The context-length refers to the train-time context available to be sampled for sparse conditioning. At test time, the models are provided the first 9 timesteps as the sequence from which the reference and conditioning frames may be then indexed for sparse conditioning to generate the next 40 timesteps such that the MSE/time is comparable over the same length. In the extreme context-length=2 case, this means that the model has never 'seen' longer range interactions, whereas when context-length=25, it sees multiscale interactions, which we ablated to consider the tradeoff  and data requirements. We will revise the description in the text to avoid ambiguity.
>
> **W5: Comparisons to deterministic PDE models.**
> Thank you for this important observation. We agree that comparisons to deterministic PDE surrogates such as FNO, WNO, DeepONet, and diffusion-based PDE models are needed to help contextualize performance beyond flow-matching baselines. To this end, we have carried out further extensive experiments. In the revised version, we extend the results to compare against pure U-Net, FNO and WNO autoregressive models as well as 3D convolutional extensions the latter two neural operators as suggested by their respective authors, which additionally convolve over time. Hyperparameter settings chosen per the original papers (Li et al. 2021, Tripura et al. 2022) for the Navier Stokes vorticity dataset, and we provide hyperparameter and additional training details in a new appendix section.
>
> These additional results confirm our hypothesis that while purpose built models such as the FNO-2D and WNO-2D (and, indeed, the U-Net) outperform TempO and other flow matching methods for next step prediction, they fail at the rollout stage with significantly higher MSE/time. We also add the baseline model with the best rollout, FNO-3D to Figure 1 to compare directly with TempO and the top flow matching competitors. A few observations: the FNO3D is the only model where the first timestep is not necessarily the most accurate, which is indicative of its treatment convolving over all 3 spatiotemporal dimensions at once; the FNO2D baseline exhibits the expected degradation as error accumulates (also indicated by the large variance seen across multiple generations) despite having the best next-step performance. We also note that Figure 1 has been modified to show the mean and standard deviation averaged across 10 randomly selected initial conditions from the withheld test set.

---

> ### Author Response · Authors · 2025-11-24
>
> | Regressor | Path   | MSE ↓     | SpectralMSE ↓ | RFNE ↓    | PSNR ↑ | Pearson ↑ | SSIM ↑ | MSE/time ↓ |
> |-----------|--------|-----------|---------------|-----------|--------|-----------|--------|------------|
> | TempO     | River  | 5.63e-02  | 3.84e-02      | 2.50e-01  | 25.19  | 0.969     | 0.786  | 2.67e-02 |
> | TempO     | Affine | 5.77e-02  | 3.98e-02      | 2.54e-01  | 25.08  | 0.968     | 0.789  | 2.91e-02 |
> | TempO     | VP     | 8.10e-02  | 5.34e-02      | 2.85e-01  | 23.61  | 0.955     | 0.731  | 2.29e-01 |
> | TempO     | VE     | 2.96e-01  | 1.73e-01      | 5.60e-01  | 17.98  | 0.821     | 0.373  | 5.02e-01 |
> | ViT       | Affine | 6.75e-02  | 4.38e-02      | 2.72e-01  | 24.40  | 0.962     | 0.758  | 8.71e-02 |
> | ViT       | River  | 6.88e-02  | 4.33e-02      | 2.73e-01  | 24.32  | 0.962     | 0.750  | 3.85e-02 |
> | ViT       | VP     | 7.77e-02  | 4.95e-02      | 2.89e-01  | 23.79  | 0.956     | 0.729  | 6.65e-02 |
> | ViT       | VE     | 1.63e+00  | 9.27e-01      | 1.35e+00  | 10.67  | 0.118     | 0.024  | 1.67e+00 |
> | U-Net     | VP     | 4.05e-01  | 3.26e-01      | 6.71e-01  | 16.62  | 0.756     | 0.323  | 2.66e-01 |
> | U-Net     | River  | 4.08e-01  | 3.28e-01      | 6.74e-01  | 16.59  | 0.752     | 0.321  | 2.79e-01 |
> | U-Net     | Affine | 4.10e-01  | 3.42e-01      | 6.76e-01  | 16.57  | 0.751     | 0.324  | 2.82e-01 |
> | U-Net     | VE     | 5.02e-01  | 3.70e-01      | 7.48e-01  | 15.68  | 0.694     | 0.263  | 2.92e-01 |
> | Baselines | FNO-2D | 6.09e-04  | 4.27e-04      | 2.54e-02  | 44.85  | 1.000     | 0.992  | 1.92e-01 |
> | Baselines | FNO-3D | 1.06e-01  | 7.34e-02      | 3.37e-01  | 22.46  | 0.945     | 0.645  | 6.37e-02 |
> | Baselines | WNO-2D | 3.72e-03  | 2.83e-03      | 6.06e-02  | 36.99  | 0.998     | 0.966  | 5.19e-01 |
> | Baselines | WNO-3D | 2.23e-01  | 1.19e-01      | 4.97e-01  | 19.21  | 0.868     | 0.452  | 2.05e-01 |
> | Baselines | U-Net  | 2.47e-03  | 1.92e-03      | 4.83e-02  | 38.772 | 0.999     | 0.976  | 1.66e+00 |
>
>
> **Minor Corrections:**
> We thank the reviewer for these helpful pointers and will incorporate citations to the recommended flow-matching-for-PDEs papers.
> We will clarify that Table 5 uses the Navier Stokes dataset, and will revise Figure 1 to plot the mean and std across 10 generated trajectories as a more nuanced comparison.
> Additionally, we will investigate the broken references, and appreciate your feedback!
>
> We thank the reviewer again for the constructive feedback and believe these clarifications and additions will significantly strengthen the paper!

---

> > ### Author Response · Authors · 2025-11-26
> >
> > As an addendum, we provide the revised Pearson correlation per Timestep graph with mean and standard deviation, and including FNO-2D and FNO-3D results to show the error accumulation and high variance of the autoregressive method past the first 10 generated steps. We additionally provide another visualised rollout to further qualify the graph, where TempO generates a rollout capturing both high frequency detail (vortices) and high fidelity to the ground truth data as opposed to the FNO-2D which diverges despite having indisputably good next-step generation.
> >
> > https://ibb.co/album/p6BNTk

---

> > > ### Comment · Reviewer_CwD2 · 2025-11-26
> > > **Thank you for the response**
> > >
> > > Dear authors,
> > >
> > > Thank you for posting a reply. I've read through it and also the feedback from other reviewers and I'm still a bit unconvinced. The benchmarks have gotten better, but they don't seem to line up with other results in the field. If I understand correctly, the Unet baseline is taken from PDEBench, but that specific implementation has known flaws and as since been improved greatly by more recent benchmark studies  (https://arxiv.org/abs/2412.00568, https://arxiv.org/abs/2209.15616, https://arxiv.org/abs/2309.01745). It also seems the main difference to prior methods is the use of conditional vs. unconditional generation, which is perhaps incremental.

---

> ### Author Response · Authors · 2025-11-27
>
> Thank you for your reply! We apologize for not clarifying the baselines in our rebuttal; we did not use the U-Net from PDEBench but rather the U-Net implementation in the PDEArena paper you linked denoted as UNet_{att}. PDEArena tests on a generated dataset using PhiFlow, with similar one-step errors reported from their UNet_{att} and ours (1.63e-3 for velocity versus 2.47e-3 for vorticity). The discrepancy in rollout error is due our much longer rollout.
>
> Regarding the novelty:
>
> Functional Flow Matching (FFM) learns vector fields in function space, but operates on static solution distributions (e.g., sampled Navier–Stokes fields) without modeling temporal evolution. We carried out additional experiments, the original FFM code even with both dense and sparse conditioning failed to produce reasonable next-step predictions, and multi-step rollouts degraded severely.
>
> TempO was designed for deterministic timeseries generation by introducing
>
>   (1) time-conditioned *latent spectral* embeddings,
>
>   (2) multi-headed attention-enhanced autoencoder for multiscale embeddings,
>
>   (3) splitting of space and time processing motivated by *PDE evolution operators*,
>
>   (4) incorporating sparse conditioning,
>
> additionally motivated by our derived theoretical approximation limits.

---

### Official Review · Reviewer_D1ut · 2025-10-30

**Soundness:** 2
**Presentation:** 3
**Contribution:** 2
**Rating:** 4
**Confidence:** 3

**Summary:**

This paper presents a method called TempO, which evolves the solution of PDEs using flow matching. First, TempO projects the input into a latent space. The evolution of the solution in latent space is trained with flow-matching, including sparse conditioning on the previous latent space values. Here, TempO uses a time-conditioned FNO to learn the latent flow.  The authors evaluate TempO against other flow-matching video-generation methods based on vision transformers and UNets in combination with different flow-matching paths. The experiments are performed on the Navier-Stokes, shallow water, and reaction-diffusion PDEs, showing that TempO reaches a lower MSE and diverges more slowly than the other models.

**Strengths:**

1. Novel combination of Flow-matching and FNO.
2. Theoretical bounds on the model approximation error are provided.
3. Extensive evaluation against other flow-matching approaches.
4. The paper is clear and well written.

**Weaknesses:**

1. No evaluation against PDE-specific models, only other flow-matching models from the video generation domain are tested. There are a number of PDE models that employ diffusion [1,2,3], for example, and also improve the rollout stability in that way.  Additionally, since the model uses an FNO in latent space, a plain FNO should also be used as a baseline.

2. In the introduction, the stochasticity of diffusion-based models is described as a disadvantage for PDEs. However, as shown in [2], the stochasticity is still helpful, even in deterministic PDEs, since it can act as an uncertainty measure. It is especially helpful for chaotic PDEs, where diffusion can help to describe the distribution of plausible trajectories (since any estimator will diverge at some point for chaotic dynamics)



[1] Serrano, L., Wang, T. X., Le Naour, E., Vittaut, J. N., & Gallinari, P. (2024). AROMA: Preserving spatial structure for latent PDE modeling with local neural fields.
[2] Lippe, P., Veeling, B., Perdikaris, P., Turner, R., & Brandstetter, J. (2023). Pde-refiner: Achieving accurate long rollouts with neural pde solvers.
[3] Holzschuh, B., Liu, Q., Kohl, G., & Thuerey, N. (2025). PDE-Transformer: Efficient and Versatile Transformers for Physics Simulations.

**Questions:**

1. For the sparse conditioning, the experiments (line 269) mention that the last 15 frames are used to condition. During inference, how does the sparse conditioning work at the beginning (ie, when you only have the embedding of the initial condition as the input)? Does the model need to encode multiple timesteps for that?

---

> ### Author Response · Authors · 2025-11-24
>
> We thank the reviewer for the positive assessment of our work, particularly highlighting the novelty of combining flow matching with FNOs, the theoretical approximation bounds, and the clarity of the writing! We appreciate the constructive feedback and address the concerns below.
>
> **W1: Missing comparisons to PDE-specific models**
> Thank you for pointing this out. We agree that comparisons to deterministic PDE surrogates such as FNO, WNO, DeepONet, and diffusion-based PDE models are needed to help contextualize performance beyond flow-matching baselines. To this end, we have carried out further extensive experiments. In the revised version, we extend the results to compare against pure U-Net, FNO and WNO autoregressive models as well as 3D convolutional extensions the latter two neural operators as suggested by their respective authors, which additionally convolve over time. Hyperparameter settings chosen per the original papers (Li et al. 2021, Tripura et al. 2022) for the Navier Stokes vorticity dataset, and we provide hyperparameter and additional training details in a new appendix section.
>
> These additional results confirm our hypothesis that while purpose built models such as the FNO-2D and WNO-2D (and, indeed, the U-Net) outperform TempO and other flow matching methods for next step prediction, they fail at the rollout stage with significantly higher MSE/time. We also add the baseline model with the best rollout, FNO-3D to Figure 1 to compare directly with TempO and the top flow matching competitors. A few observations: the FNO3D is the only model where the first timestep is not necessarily the most accurate, which is indicative of its treatment convolving over all 3 spatiotemporal dimensions at once; the FNO2D baseline exhibits the expected degradation as error accumulates (also indicated by the large variance seen across multiple generations) despite having the best next-step performance. We also note that Figure 1 has been modified to show the mean and standard deviation averaged across 10 randomly selected initial conditions from the withheld test set.
>
> | Regressor | Path   | MSE ↓     | SpectralMSE ↓ | RFNE ↓    | PSNR ↑ | Pearson ↑ | SSIM ↑ | MSE/time ↓ |
> |-----------|--------|-----------|---------------|-----------|--------|-----------|--------|------------|
> | TempO     | River  | 5.63e-02  | 3.84e-02      | 2.50e-01  | 25.19  | 0.969     | 0.786  | 2.67e-02 |
> | TempO     | Affine | 5.77e-02  | 3.98e-02      | 2.54e-01  | 25.08  | 0.968     | 0.789  | 2.91e-02 |
> | TempO     | VP     | 8.10e-02  | 5.34e-02      | 2.85e-01  | 23.61  | 0.955     | 0.731  | 2.29e-01 |
> | TempO     | VE     | 2.96e-01  | 1.73e-01      | 5.60e-01  | 17.98  | 0.821     | 0.373  | 5.02e-01 |
> | ViT       | Affine | 6.75e-02  | 4.38e-02      | 2.72e-01  | 24.40  | 0.962     | 0.758  | 8.71e-02 |
> | ViT       | River  | 6.88e-02  | 4.33e-02      | 2.73e-01  | 24.32  | 0.962     | 0.750  | 3.85e-02 |
> | ViT       | VP     | 7.77e-02  | 4.95e-02      | 2.89e-01  | 23.79  | 0.956     | 0.729  | 6.65e-02 |
> | ViT       | VE     | 1.63e+00  | 9.27e-01      | 1.35e+00  | 10.67  | 0.118     | 0.024  | 1.67e+00 |
> | U-Net     | VP     | 4.05e-01  | 3.26e-01      | 6.71e-01  | 16.62  | 0.756     | 0.323  | 2.66e-01 |
> | U-Net     | River  | 4.08e-01  | 3.28e-01      | 6.74e-01  | 16.59  | 0.752     | 0.321  | 2.79e-01 |
> | U-Net     | Affine | 4.10e-01  | 3.42e-01      | 6.76e-01  | 16.57  | 0.751     | 0.324  | 2.82e-01 |
> | U-Net     | VE     | 5.02e-01  | 3.70e-01      | 7.48e-01  | 15.68  | 0.694     | 0.263  | 2.92e-01 |
> | Baselines | FNO-2D | 6.09e-04  | 4.27e-04      | 2.54e-02  | 44.85  | 1.000     | 0.992  | 1.92e-01 |
> | Baselines | FNO-3D | 1.06e-01  | 7.34e-02      | 3.37e-01  | 22.46  | 0.945     | 0.645  | 6.37e-02 |
> | Baselines | WNO-2D | 3.72e-03  | 2.83e-03      | 6.06e-02  | 36.99  | 0.998     | 0.966  | 5.19e-01 |
> | Baselines | WNO-3D | 2.23e-01  | 1.19e-01      | 4.97e-01  | 19.21  | 0.868     | 0.452  | 2.05e-01 |
> | Baselines | U-Net  | 2.47e-03  | 1.92e-03      | 4.83e-02  | 38.772 | 0.999     | 0.976  | 1.66e+00 |

---

> > ### Author Response · Authors · 2025-11-26
> >
> > As an addendum, we provide the revised Pearson correlation per Timestep graph with mean and standard deviation, and including FNO-2D and FNO-3D results to show the error accumulation and high variance of the autoregressive method past the first 10 generated steps. We additionally provide another visualised rollout to further qualify the graph, where TempO generates a rollout capturing both high frequency detail (vortices) and high fidelity to the ground truth data as opposed to the FNO-2D which diverges despite having indisputably good next-step generation.
> >
> > https://ibb.co/album/p6BNTk

---

> > > ### Comment · Reviewer_D1ut · 2025-11-27
> > >
> > > Thank you for the clarifications and the FNO and UNet baseline experiments.
> > >
> > > However, I agree with reviewer CwD2 that the chosen additional baselines might be too weak. Additionally, it would have been good to compare against one of the mentioned, much more recent methods that apply diffusion (also in the latent space, such as AROMA) to the problem.  These methods were specifically designed to enhance rollout stability. Since you also argued about the computational overhead of these diffusion methods, an experiment showing this advantage of flow-matching would have been beneficial.

---

> > > > ### Author Response · Authors · 2025-12-02
> > > >
> > > > We thank the reviewer for their thoughtful and detailed feedback which highlights the novelty of TempO's combination of latent flow matching with FNO, the provided theoretical bounds, and the extensive evaluation against other flow-matching approaches.
> > > >
> > > > To clarify our additional baselines in the rebuttal, we indeed use the competitive attention-based U-Net implementation adapted from the DDPM paper that Reviewer CwD2 linked, including attention, which in the PDEArena is denoted as UNet_{att}. We find that though they generate a separate dataset using PhiFlow, the one-step error reported from their UNet_{att} is reasonably similar (1.63e-3 for velocity versus 2.47e-3 for vorticity). The discrepancy in rollout error is due our much longer rollout.
> > > >
> > > > We appreciate the suggestion to compare with latent diffusion-based models! We configured and ran AROMA on the Navier–Stokes dataset (without truncating the time series, as is done in AROMA) for a comparable number of epochs to our training. The metrics were far from the AROMA paper despite using their own config file, likely because the autoencoder did not converge within the 1000 epochs. In these preliminary tests (1000 epochs of the AROMA autoencoder and 500 epochs for the AROMA model), we observed poor convergence and a considerable reduction in efficiency compared to our model.
> > > >
> > > > **Training cost comparison (Navier–Stokes dataset) on a single GPU:**
> > > >
> > > > | Model                  | Total epochs (per config) | Time per 100 epochs (hours)          |
> > > > |------------------------|-------------|---------------------------------|
> > > > | AROMA encoder–decoder  | 10000       | 5.34                            |
> > > > | AROMA model            | 500         | 6.14                            |
> > > > | TempO encoder–decoder  | 500         | 0.70   |
> > > > | TempO                  | 100         | 1.18                            |
> > > >
> > > > Overall, reproducing AROMA is significantly more expensive and less stable than our approach. TempO, by comparison, show strong convergence and computational efficiency as confirmed by our broad set of experiments against state of the art flow matching models. We furthermore added significant baseline experiments conducted during the rebuttal, demonstrating its crucial rollout stability.

---

> ### Author Response · Authors · 2025-11-24
>
> **W2: Discussion of stochastic diffusion in deterministic PDEs.**
> We appreciate this nuanced point. Indeed, works such as PDE-Refiner show that stochasticity can indeed be valuable even for deterministic PDEs, e.g. by providing a natural mechanism for uncertainty quantification or by representing ensembles of plausible trajectories in chaotic systems. We fully agree with this perspective. Our discussion in the introduction was not intended to argue that stochasticity is unhelpful, but rather that it introduces additional computational overhead and variance when the primary goal is operator learning of a deterministic map. Even for chaotic PDEs, the operator mapping from parameters/initial conditions to short-term evolution is deterministic. The stochasticity is useful for long-term uncertainty propagation but not required for the operator learner itself. TempO therefore offers an alternative: a deterministic flow formulation that preserves the efficiency and stability benefits of operator learning while still allowing, if desired, the introduction of controlled stochasticity at prediction time (e.g., via noise perturbations of the initial condition or noise augmentation paths).
>
> We revise the introduction to be more nuanced about this claim:
> "Other existing efforts which leverage diffusion (Molinaro et al., 2025; Yao et al., 2025; Huang et al., 2024) move towaard more natural representations and, in some settings, benefit from the stochasticity inherent to diffusion (e.g. for uncertainty quantification or modeling ensembles of plausible trajectories in chaotic regimes) (Lippe et al., 2023, Serrano et al., 2024, Holzschuh et al., 2025). However, their stochastic sampling procedures can introduce computational overhead and variance when the goal is to learn a sharp, deterministic operator. Flow matching provides a complementary alternative: vector field regression aligns naturally with learning PDE operators, which themselves describe deterministic time derivatives, and yields efficient ODE-based sampling without iterative denoising."
>
> **Q1: Sparse conditioning during inference.**
> We thank the reviewer for raising this important clarification. During inference, we follow the precedence of both Davtyan et al. 2023 and Li et al. 2021
> % (RIVER, which introduces the sparse conditioning, and the prototypical FNO paper, respectively)
> which provide a 'leadup' of $n$ timesteps, where $n>=1$ (though ideally, $n>2$ to allow for unique reference and conditioning frames. Currently, the first 9 timesteps are available either as a dense leadup (for the pure FNO), or as the sequence from which the reference and conditioning frames may be then indexed for sparse conditioning to generate the next 40 timesteps (for the flow matching models). We also draw attention to the impact that this leadup has on the generation, from $n=2$ to $n=25$, in the ablation shown in Table 5.
>
> We revise the paragraph describing this experiment to be more clear:
> "Models follow the inference protocol used by (Davtyan et al., 2023; Li et al., 2021):: a short lead-up of 9 initial frames is provided, with the baseline FNO using all 9, and the sparsely conditioned flow-matching models using only two frames in the lead-up as conditioning and reference frames. The conditioning frame is then pinned and the temporal offset vector is incremented while the reference frame is set to timestep $t-1$ to generate timestep $t$."
>
> Overall, we appreciate the feedback especially in clarifying the messaging of our paper to be more nuanced and placed within the wider discussion toward PDE rollouts.

---

### Author Response · Authors · 2025-12-03
**Summary Comment**

We thank the reviewers for their thorough and insightful feedback. We have precisely addressed all reviewer comments, including (i) running the additional experiments they requested, (ii) providing further clarification or correction where misunderstandings may have arisen, and (iii) substantially expanding both our empirical evaluation and methodological exposition. Their suggestions resulted in:

* *Comparison to base neural operators:* Reviewers 1, 2, and 3 noted that comparisons against vanilla FNO, UNet, and other operator learning models would provide additional context. We have performed a comprehensive and targeted expansion of our experimental evaluation, addressing every reviewer request with new baselines (FNO-2D, FNO-3D, WNO-2D, WNO-3D, UNet) and recent related works (FFM, AROMA) which complement our backbone variants, component ablations, flow-matching comparisons, long-horizon rollouts, and spectral-domain analyses. We demonstrate how TempO outperforms both standard autoregressive and 3D-convolutional FNO and UNet baselines, particularly in long-horizon rollout accuracy, without losing high frequency detail. It requires an order of magnitude fewer epochs to train than AROMA (Serrano et al. 2024), a latent diffusion-based model built for PDE rollouts. Though at a glance similar to the Functional Flow Matching (FFM, Kerrigan et al. 2023) which was designed for learning static solution distributions, the original FFM code even with both dense and sparse conditioning failed to produce reasonable next-step predictions, and multi-step rollouts degraded severely.

* *Clarity and novelty:* To address reviewer concerns (R2, R3), we clarified the introduction and methods section to explicitly highlight the novelty of TempO:

  1. Time-conditioned spectral latent embeddings,
  2. Multi-headed attention-enhanced autoencoder for multiscale embeddings,
  3. Splitting of space and time processing via channel folding motivated by PDE evolution operators,
  4. Incorporating sparse conditioning

* *Experiment details:* We have clarified the experiment's and ablation's method, and emphasized the clear distinction between TempO and previous methods, including FFM and Conditional Flow Matching.

We are confident these revisions support the strengths of our proposed method TempO. TempO introduces a novel operator-based flow matching framework that achieves stable, high-fidelity long-horizon PDE rollouts while remaining lightweight and efficient. Its theoretical guarantees and strong empirical results demonstrate clear advantages over existing autoregressive and neural-operator baselines.

In particular, (i) Reviewers 1, 3, and 4 praise the novel combination of latent Flow Matching with FNO, and the theoretical analysis establishing approximation bounds. (2) Extensive evaluation against multiple flow-matching approaches, noise schedules, and backbones is recognized (R1, R2, R4), demonstrating both stability and efficiency in long-horizon PDE forecasting. (3) TempO achieves highly stable 40-step autoregressive rollouts with strong spectral and parameter efficiency (R3, R4).(4) The manuscript is clear and well-motivated, with reviewers noting the alignment of FNO’s spectral bias with continuous PDE dynamics (R1, R3, R4).

TempO delivers the first principled integration of latent flow matching with neural operators mimicking classical PDE evolution operators, offering an efficient, stable and spectrally accurate alternative to autoregressive and diffusion-based PDE forecasting. Our theory explains why operator-based flow matching is more efficient and accumulates error more slowly, and our extensive experiments confirm consistent long-horizon gains across diverse PDEs. Together, this establishes TempO as a robust and efficient foundation for high-fidelity scientific forecasting.

---

### Meta-Review · Area_Chair_ZK4N · 2026-01-05

**Summary:**

The paper addresses the long-horizon forecasting problem for data-driven PDE surrogates, which is affected by error accumulation in autoregressive (AR) models. The authors propose a flow-matching formulation that models the temporal evolution of the dynamics directly in a latent space, predicting the state representation at arbitrary query times. This avoids autoregressive rollouts and mitigates error accumulation. The flow regressor is implemented using a time-conditioned Fourier Neural Operator, and the model employs channel folding to enable efficient processing. The paper provides theoretical justification for the proposed formulation and presents comparisons with alternative flow-matching baselines on three datasets.

The reviewers acknowledge the sound theoretical grounding, the extensive evaluation against alternative flow-matching approaches, the consistent performance gains, and the novel combination of Fourier Neural Operators with flow matching. They nonetheless raise several concerns, including the lack of comparisons with deterministic baselines and recent state-of-the-art generative models, the need for additional evaluations on longer rollouts and more challenging benchmarks, missing technical details, and a clearer positioning with respect to closely related work.

**Reviewer Concerns:**

The authors responded with a strong rebuttal, adding new experiments, providing the requested technical clarifications, and elaborating on the motivation and positioning of their approach within the existing literature. The reviewers acknowledge these clarifications and the additional experimental results. However, some remain unconvinced regarding the level of originality and suggest that comparisons with recent state-of-the-art latent-space generative models would further strengthen the contribution.

Overall, I believe the authors have provided a thorough and constructive rebuttal, with well-articulated arguments supporting their methodological choices. A revision of the paper that more clearly highlights the originality and positioning of the approach, together with comparisons against the suggested state-of-the-art generative models, would significantly strengthen the submission. It is also worth noting that a substantial body of related work exists in the video generation literature comparing autoregressive and direct generative forecasting paradigms, which could provide additional support for the proposed approach.

**Reviewer Scores:**

RD1ut, rating 4, indicates that the baselines are still too weak and comparison with more recent methods are needed, would probably not increase their score

RCwD2, rating 4, indicates that they are not fully convinced by the rebuttal, would probably not increase their score

RtxKP, rating 2, excellent rebutal, according to me should increase their score, but considering their arguments not sure he would do

Rsseo, rating 6, would probably keep their score

---

### Decision · Program_Chairs · 2026-01-26

Reject